**communications**

**biology**

# Mapping density, diversity and species-richness of the Amazon tree flora

Using 2.046 botanically-inventoried tree plots across the largest tropical forest on Earth, we mapped tree species-diversity and tree species-richness at 0.1-degree resolution, and investigated drivers for diversity and richness. Using only location, stratified by forest type, as predictor, our spatial model, to the best of our knowledge, provides the most accurate map of tree diversity in Amazonia to date, explaining approximately 70% of the tree diversity and species-richness. Large soil-forest combinations determine a significant percentage of the variation in tree species-richness and tree alpha-diversity in Amazonian forest-plots. We suggest that the size and fragmentation of these systems drive their large-scale diversity patterns and hence local diversity. A model not using location but cumulative water deficit, tree density, and temperature seasonality explains 47% of the tree species-richness in the terra-firme forest in Amazonia. Over large areas across Amazonia, residuals of this relationship are small and poorly spatially structured, suggesting that much of the residual variation may be local. The Guyana Shield area has consistently negative residuals, showing that this area has lower tree species-richness than expected by our models. We provide extensive plot meta-data, including tree density, tree alpha-diversity and tree species-richness results and gridded maps at 0.1-degree resolution.

Tree alpha-diversity (here defined as Fisher's alpha[1] of a tree-inventory plot) in Amazonia is influenced by regional and local drivers[2,3]. Regional drivers include large-scale patterns in rainfall[4–6], temperature or for instance large-scale gradients in soil fertility[5]. In contrast, local drivers include local differences in soil type[2,7,8], soil hydrology[9], flooding[10] and small-scale ecological processes, such as disturbance[11,12], or frequency-dependent mortality[13–15]. Fisher's alpha provides information on both species-richness in a sample of known size and the relative abundances of all species in that sample, providing both aspects of biodiversity. Species-richness is an important aspect of biodiversity, however, and often easier to communicate. In this paper we will use both indices.

Forest types in Amazonia are to a large extent driven by soil and hydrology (e.g. groundwater, seasonal flooding or waterlogging)[8,9,16] and significant differences in composition and tree alpha-diversity between forest types exist[2,3]. Terra-firme (see methods section for a definition of all forest types) is the forest type with the highest tree alpha-diversity, while forests on white sand, várzea, igapó and swamps generally have lower tree alpha-diversity[2] and higher dominance[17], which are inversely related to each other. Lower tree alpha-diversity has been attributed directly to poor nutrient status in white sand forests and flooding and waterlogging in várzea, igapó and swamps. Assuming that poor soils and flooding require traits that by means of a trade-off reduce fitness on terra-firme[18,19], it could be argued that in line with the Island theory of Biogeography [20], the relative area of any particular forest type is an important driver of large-scale diversity[21] and thus influences local diversity[7,17,20,22,23]. Following this reasoning, forest types such as white-sand forest and swamp forest have small areas and a fragmented distribution across Amazonia, resulting in relatively small meta-populations and hence a lower potential alpha-diversity (subsequently influenced by local processes). In essence, their diversity is therefore likely strongly influenced by local extinctions and limited migration from a smaller meta-species pool[23,24].

In 2000 and 2003 the Amazon Tree Diversity Network (ATDN) published its first maps of tree diversity of Amazonia at 1-degree resolution, based on a dataset of 268 and 425 terra-firme sampling plots, respectively[25,26]. At the time, plots in other forest types were omitted as they had lower diversity and were small in numbers. In a subsequent attempt, an interpolation of all plots (725) was used. It was assumed that an interpolation of all plot data would create the regional signal, while the residuals from the spatial interpolation could be regarded as the local ecological signal, including forest type and residual error[2,3].

After a period of 20 years, sufficient plots are now available to map diversity directly by main forest type. Here, we produce a new tree diversity map of the original forest cover of Amazonia at 0.1-degree resolution, using soil information of all plots and a large-scale soil map[27,28]. We constructed four spatial models for tree alpha-diversity, one for terra-firme, one for white sand forest, one for floodplain forests (várzea, igapó), and one for swamp forests. Results were subsequently mapped on the major soil types at 0.1-degree resolution. Afterwards, the same approach was used to map tree density and tree species-richness across Amazonia.

Although a considerable number of plots has been added to the general database, we recognize there are several caveats in this attempt. First, it has long been recognised that personal experience and training can significantly affect the identifications (including distinguishing among similar species) of trees and thus the richness of a plot[29], and that with a large network of over 230 contributors this might create a bias but one for which we cannot easily correct. Plots with clearly poor identification are, however, not included in the ATDN database. Second, the level of correct identification of specimens in herbaria may differ considerably[30].

Third, the spatial differences in the number of specimens collected over Amazonia are large[31–33]. Arguably, species identification would be better, and therefore richness higher, all other things being equal, in areas where collecting intensity is high, and large, active herbaria are present (e.g. Manaus, Belem, Cayenne). Finally, time of establishment may also affect plot tree species-richness. Plots of the ATDN were established between 1934 and 2020. The number of known (described) species for Amazonia has increased considerably in that period, especially since the 1940s[31]. In addition, new taxonomic insights contribute to splitting dominant taxa into separated lineages[34,35] or merging rare species into common ones[31]. It should be expected therefore that, all other things being equal, early plots have fewer species than plots established more recently. The effect of this would depend on the quality of the botanists identifying the species, as they could either force a tree into a known species or keep it as a morpho-species, in which case there would be less effect on tree alpha-diversity and tree species-richness.

We tested potential drivers for the observed patterns in tree species-richness, including climatic variables, soil variables, time of establishment, and intensity density. Our results show that area has a strong effect on tree diversity and richness, as do cumulative water deficit and tree density. Finally, we produce a full list of all plots with metadata, tree density, alpha-diversity, species-richness data, and plot references, and two A3 maps with our main map results

## Results

**Tree density**. Tree density (the number of trees/ha) for the plots ranged from 80–2629 trees/ha (Supplementary Fig. 1a), with an average density of 563 trees/ha based on 1956 plots (after removing outlier plots with density <200 and >900 trees/ha), with a total number of over a million trees. Density differed little between forest types, with slightly higher densities in swamp, white-sand forest, and terra firme forest of the Guyana Shield (Supplementary Fig. 1b). The highest tree densities were found in north-western Amazonia. Lowest tree densities were found in central-south, and eastern Amazonia (Supplementary Fig. 1c, Note that Supplementary Fig. 6c, is in fact the regional tree density). The spatial model explained 43% of the variation in tree density across Amazonia (Supplementary Fig. 1d), of which the residuals had no significant spatial autocorrelation (Supplementary Fig. 1d).

**Tree alpha-diversity**. Tree alpha-diversity (defined here as Fisher's alpha) ranged from 0.51 to 257 per plot, with an average of 56. Data showed a strongly skewed distribution (Fig. 1A). Forest type explained a significant amount of variation of Fisher's alpha ($R^2 = 23\%$, $p < 0.001$, Fig. 1B). The highest tree alpha-diversity was found in the terra-firme forests of the Pebas region (western Amazonia) and French Guiana (Fig. 1C, D). Lowest tree alpha-diversity was found on the sandy soils of Guyana, Upper Rio Negro, southern Amazonia and floodplains along the rivers and swamps. The combined spatial model explained 66% of the variation in tree alpha-diversity across Amazonia (Fig. 1D), and its residuals showed no significant spatial autocorrelation (Moran's I < 0.001 n.s.). The standard error for Fisher's alpha was mostly low (Supplementary Fig. 2a) and consistent across regions (Supplementary Fig. 2b) although higher for white sand forest and swamp forests (Supplementary Fig. 2c), resulting in higher standard errors in the white sand areas of the Upper Rio Negro and Guianas. Residuals of the combined spatial model had a mean close to zero and did not differ much between forest type and region, showing no spatial pattern, consistent with a non-significant Moran's I (Supplementary Fig. 3).

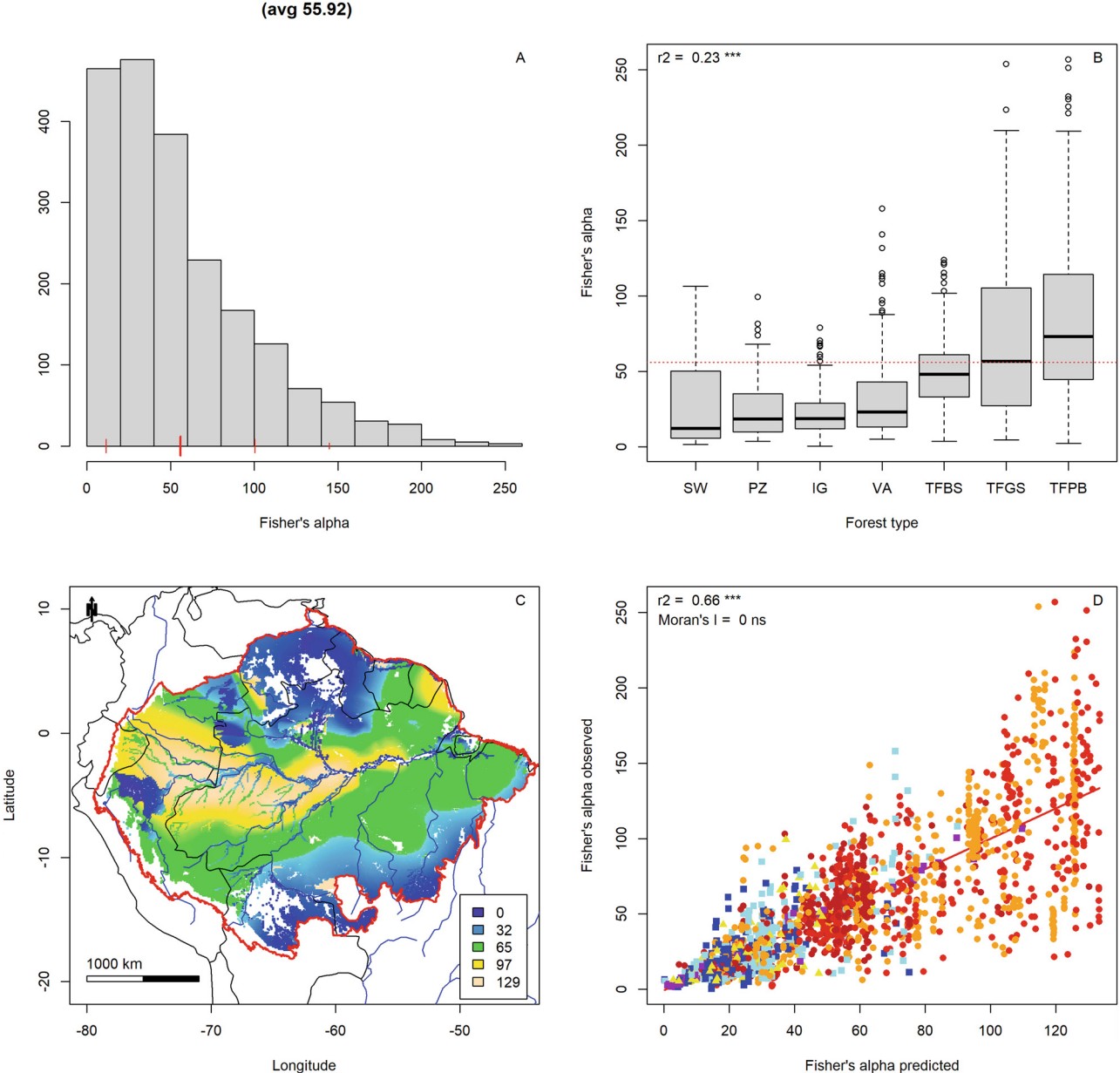

**Fig. 1 Tree alpha-diversity (Fisher's alpha) in Amazonia. A** Histogram of tree alpha-diversity in 2046 ATDN plots. Red lines, mean and mean ± 2 sd. **B** Tree alpha-diversity by major forest type. **C** Map of tree alpha-diversity across Amazonia. Legend truncated at 0 and mean + 2 standard deviation of the mean. Amazonian Biome limit - red[79]. **D** Observed values of tree diversity vs modelled values of tree diversity on the 2046 plots used for mapping. The significance or Moran's I was tested with the function Moran.I() of ape[61]. Marker colours: Red: Terra Firme Pebas Formation; Brown: Terra Firme Brazilian Shield; Orange: Terra Firme Guyana Shield; Yellow: White sand forest; Light blue: Varzea; Dark blue: Igapo; Purple: Swamp. Map created with custom R[80] script. Base map source (country.shp, rivers.shp): ESRI (http://www.esri.com/data/basemaps, © Esri, DeLorme Publishing Company).

Mapping the logarithm of Fisher's alpha, to account for its skewed distribution (Supplementary Fig. 4a), did not produce a very different spatial pattern (Supplementary Fig. 4c) but a slightly better model (Supplementary Fig. 4c, $R^2 = 73\%$). As Fisher's alpha is the more commonly used metric, we have kept this version in the main text.

Using only location (not stratified by forest-soil)[2,3], provided a map with a comparable overall regional pattern, but with much more average values as nearby low and high diversity plots of different forest types were mixed in the local estimation (Supplementary Fig. 5a). This model explained 45% of local Fisher's alpha (Supplementary Fig. 5b). Part of the local signal,

following[2,3], was explained by forest type ($r^2 = 19\%$, Supplementary Fig. 5c). Thus, the total explained variation by adding the two models would be 45% + 19% of 100−45% = 55.5%, which was 10% less than the spatial model with forest type included. Regions had no more effect on the residuals of the spatial model (Supplementary Fig. 5d).

**Tree species-richness.** Tree species-richness, defined as the number of species per ha, ranged from 3 to 357 with an average of 121 species/ha. The data were less skewed than those of Fisher's alpha (Fig. 2A). Forest type explained a significant amount of

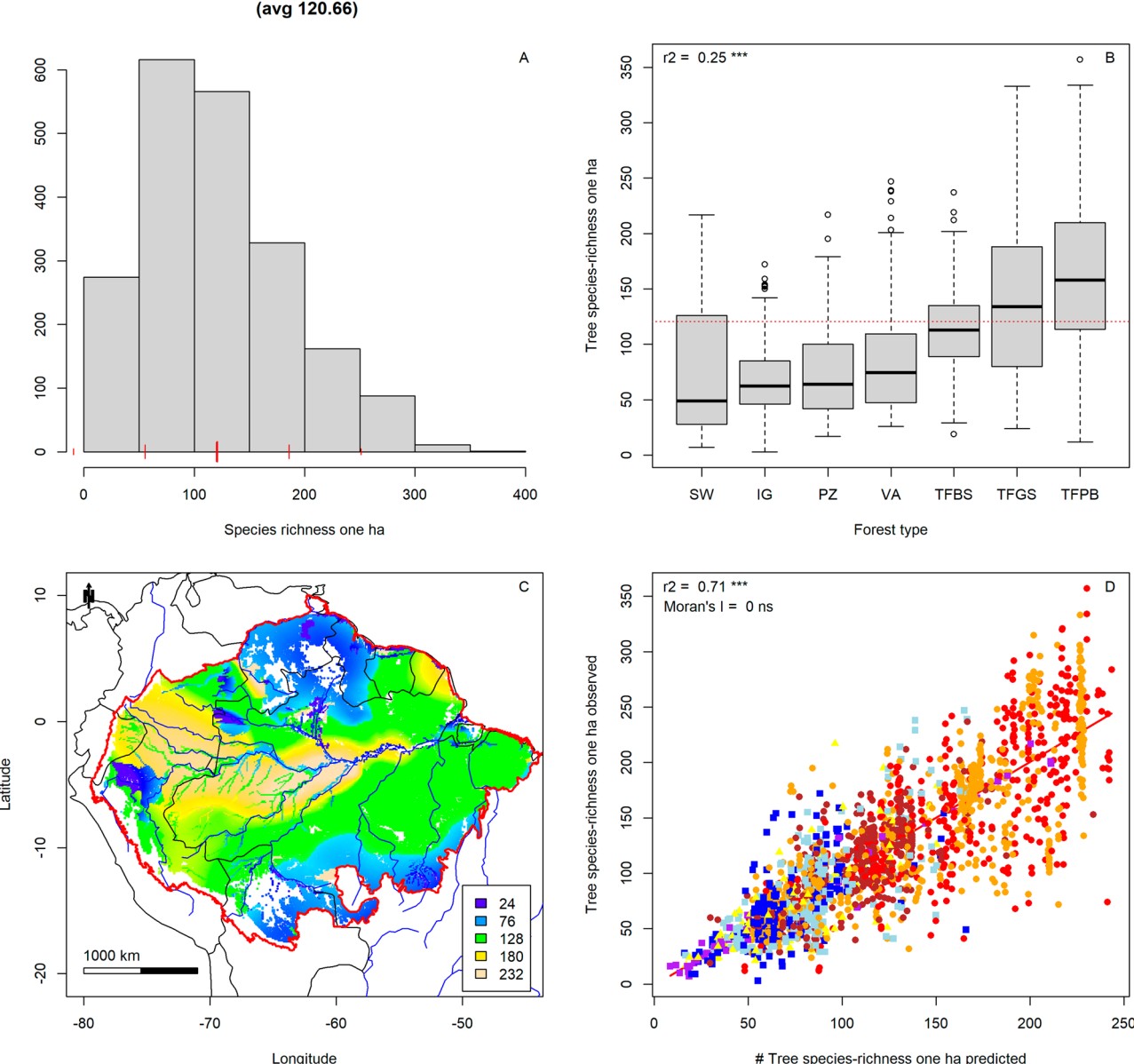

**Fig. 2 Tree species-richness (species/ha) in Amazonia. A** Histogram of tree species-richness in 2046 ATDN plots. **B** Tree species-richness by major forest type. **C** Map of tree species-richness across Amazonia. Legend truncated at mean ± 2 standard deviations of the mean. Amazonian Biome limit - red[79]. **D** Observed values of tree species-richness vs modelled values of tree species-richness on the 2046 plots used for mapping. The significance or Moran's I was tested with the function Moran.I() of ape[61]. Marker colours: Red: Terra Firme Pebas Formation; Brown: Terra Firme Brazilian Shield; Orange: Terra Firme Guyana Shield; Yellow: White sand forest; Light blue: Varzea; Dark blue: Igapo; Purple: Swamp. Map created with custom R[80] script. Base map source (country.shp, rivers.shp): ESRI (http://www.esri.com/data/basemaps, © Esri, DeLorme Publishing Company).

variation in species-richness ($R^2 = 25\%$, $p < 0.001$) (Fig. 2B). The highest species-richness was found in the terra-firme forests of the Pebas region (western Amazonia) and French Guiana (Fig. 2C). Lowest species-richness was found on the sandy soils of Guyana, Upper Rio Negro, southern Amazonia, and on flood-plains along the rivers and swamps. The combined spatial model explained 71% of the variation in tree species-richness across Amazonia (Fig. 2D), and its residuals showed no significant spatial autocorrelation (Moran's I < 0.001 n.s.).

The standard error for tree species-richness was mostly low (Supplementary Fig. 6a) and rather constant across regions (Supplementary Fig. 6b) but higher for white sand forest and swamp forests (Supplementary Fig. 6c), resulting in higher standard errors in the white sand areas of the Upper Rio Negro and Guianas (Supplementary Fig. 6d). Residuals of the combined

spatial model had a mean of close to zero and did not differ much between forest type and region (Supplementary Fig. 7). Mapping the logarithm of species richness, to account for its slightly skewed distribution (Supplementary Fig. 8), did not produce different results. Species-richness in 500 individuals showed identical patterns as species-richness/ha (Figs. S9–S11).

Mapping the species richness map 2046 times, leaving each plot out once and estimating its value with the map it did not contribute to, provided a final test of our model. If all data is involved the $R^2$ of the observed tree species-richness vs. the predicted tree species-richness dropped from 71 to 65%. This reduction is mainly caused by the plots on white sand (partial $R^2$ dropped from 56 to 16%) and swamp forest (96%–16%—see the close alignment of the swamp plots to the regression line in Figs. 1D, 2D), where sample sizes are much smaller. Because it

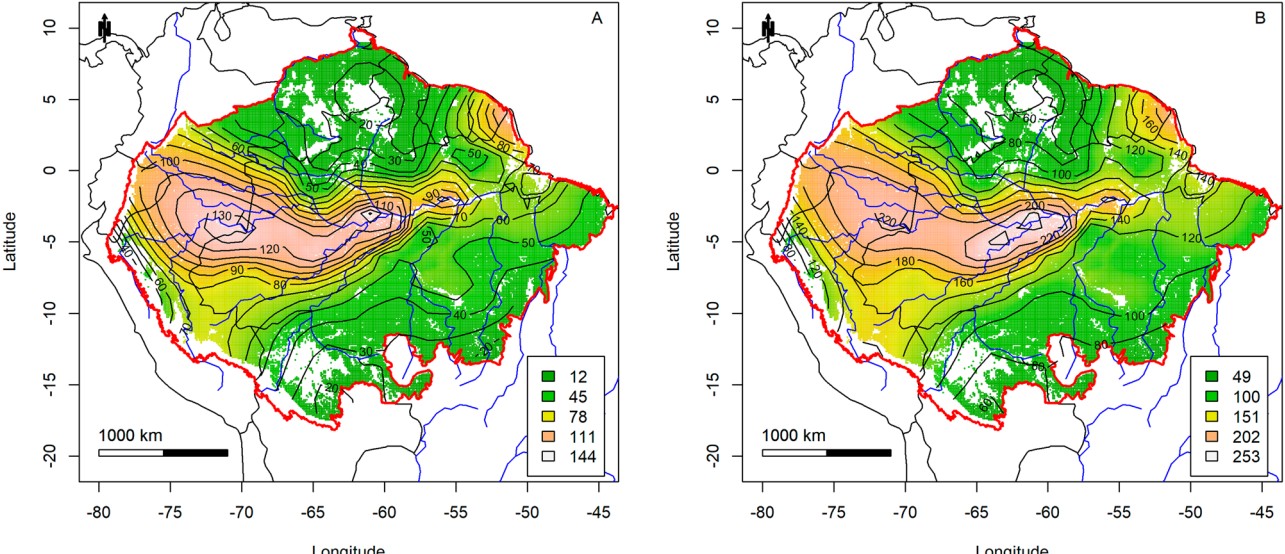

**Fig. 3 Tree alpha-diversity and tree species-richness of terra-firme forest in Amazonia. A** Map of interpolated tree alpha-diversity (Fisher's alpha), based on 1441 terra firme plots. **B** Map of tree species-richness (number of species/ha by plot), based on 1441 terra firme plots. Red polygon: Amazonian Biome limit[79]. Map created with custom R[80] script. Base map source (country.shp, rivers.shp): ESRI (http://www.esri.com/data/basemaps, © Esri, DeLorme Publishing Company).

was not possible to produce a map based on the leave one out principle our final maps are based on a model where all data is used (Figs. 1C, 2C).

**The effects of abiotic factors on tree diversity and richness.** As most plots were established in terra-firme forest, this forest type was used as an example to study the effects of climate, soil, collecting and year of establishment on tree alpha-diversity and richness across Amazonia. Tree species diversity and richness per ha of terra-firme forest had a clear maximum in central Amazonia, where the interpolation model predicted over 250 species per ha (Fig. 3).

The spatial model of Fisher's alpha explained 59.0% of the variation of plot level Fisher's alpha, while the spatial model for species richness explained 65.5% of the variation of species-richness at plot level. Tree density (at plot level) affected species-richness (Supplementary Fig. 12b $R^2 = 0.16^{***}$). Regional tree density explained tree species-richness/ha even better (Supplementary Fig. 13, $R^2 = 0.26^{***}$). As sample size (N) affects tree species-richness, the regional richness pattern is better viewed with the average regional tree density pattern and for a sample of similar size ($n = 500$). However, regional tree density explained tree species-richness per 500 individuals less strongly (Supplementary Fig. 14, $R^2 = 0.16^{***}$). Plot tree density only explained 5% of the variation in tree species-richness in a sample of 500 individuals. Supplementary Table 1 shows the results of all models used.

Cumulative water deficit had a negative effect on tree species-richness (Supplementary Fig. 15) with a decrease of 17 species for each 100 mm deficit ($r^2 = 27\%^{***}$) and a loss of maximum richness of 25 species per 100 mm deficit. Central Amazonia had more species per ha than expected based on cumulative water deficit alone (Fig S15d). In contrast, southern Amazonia and especially the Guyana Shield had a much lower richness than expected based on cumulative water deficit alone.

Annual rainfall (Bioclim 12) had a positive effect on tree species-richness in terra-firme forest ($R^2 = 22\%^{***}$), both for the mean and the upper limit (Supplementary Fig. 16b). An increase of 1000 mm of annual precipitation resulted in an increase of

50 species per ha on average, and 80 species maximum. Residuals suggest that central Amazonia has 70–125 species per ha more than would be expected based on annual rainfall (Supplementary Fig. 16d). Northern Amazonia (defined here as the Guyana Shield) and southern Amazonia have less than 30 species per ha than expected based on annual rainfall.

Soil pH had a small effect on tree species-richness (Supplementary Fig. 17); Sum of bases had no significant effect with quantile regression (Supplementary Fig. 18) and a very small, significant effect with least-squares regression (0.3%, Supplementary Table 1); whether a plot was situated on the Pebas formations or cratons (Guyana or Brazilian Shield areas, Supplementary Fig. 19) also explained little variation (5%).

Collecting intensity explained 13% of species-richness on the plots (Supplementary Fig. 20). Especially in the Manaus area, the residuals of this relationship were very high (Supplementary Fig. 20d).

Most plots (78%) were established after 2000 (Supplementary Fig. 21a, b). Establishment year, however, had no effect on the richness across the full dataset (Fig. S21b), but plots established before 1980 were primarily found in the Guianas and eastern Amazonia (Supplementary Fig. 21c). Only after 1980 the distribution was more evenly spread across Amazonia. There was a very small (but significant) effect on plot tree species-richness per ha for plots established before and after 1980 (Supplementary Fig. 21d).

As cumulative water deficit, regional tree density and collecting intensity all had significant effect on tree species-richness, we combined these factors in models: alone these variables explained the following proportions (Supplementary Table 1): cumulative water deficit 27%, regional tree density (RD) 26% and collecting intensity (CI) 13%; combined cumulative water deficit+D 38%; cumulative water deficit+CI 28%; D + CI 29%. cumulative water deficit+D + CI 38% (Supplementary Fig. 22). Thus, collecting intensity contributed little to a model with two or three variables (Supplementary Table 1). Similar results were obtained by combining cumulative water deficit, tree density and location in Pebas formation (Supplementary Fig. 23). Adding other soil variables to these models contributed nothing to the explained variation.

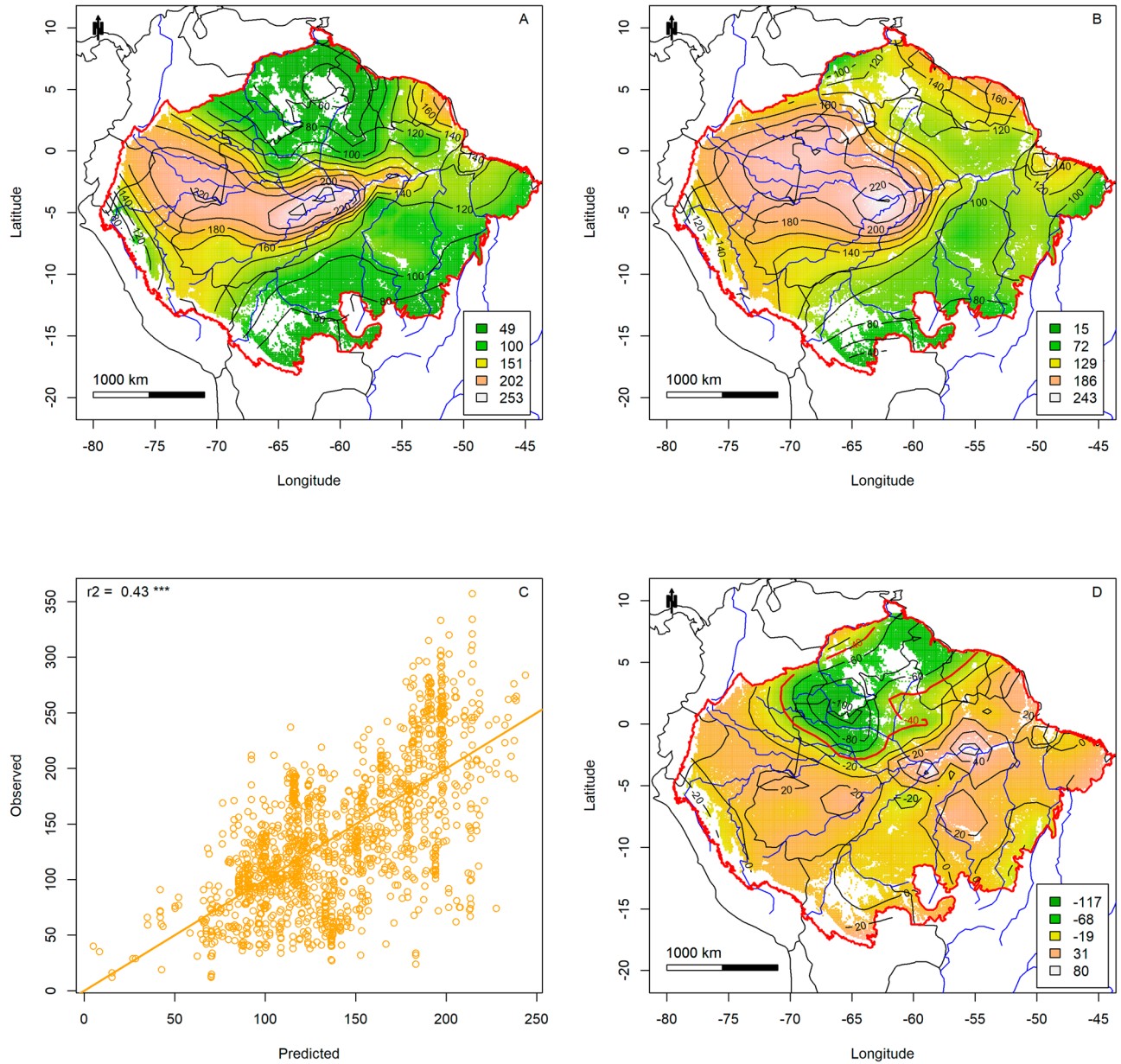

**Fig. 4 The effect of cumulative water deficit (mm), tree density, and temperature seasonality on tree species-richness. A** Tree species-richness observed. **B** Tree species-richness as predicted by cumulative water deficit, regional tree density, and temperature seasonality. **C** Model performance, showing predicted and observed tree species-richness. **D** Residuals of tree species-richness predicted by cumulative water deficit, regional tree density, and temperature seasonality (**A**, **B**). All figures based on 1441 terra firme plots. Amazonian Biome limit - red[79]. Map created with custom R[80] script. Base map source (country.shp, rivers.shp): ESRI (http://www.esri.com/data/basemaps, © Esri, DeLorme Publishing Company).

The combination of cumulative water deficit, regional tree density and temperature seasonality (Bioclim 04) explained 43% of the tree species-richness per ha (Fig. 4). In the predictions of this model, the 'dry transverse belt'[36] across Amazonia was clearly visible (Fig. 4B). Adding collecting intensity to this model did not improve its explained variation (Supplementary Fig. 24). The residuals of this model showed a clear spatial pattern with low to moderate positive residuals in most of Amazonia and one area (upper Rio Negro – Guianas) with high negative residuals (indicated by a red line in Fig. 4D).

## Discussion

Major soil/forest type exerts a significant, strong influence on tree alpha-diversity and tree species-richness in Amazonia (Figs. 1, 2).

Stratifying plots into four major soil and forest-type combinations and mapping for each major type separately allowed us to map Amazonian tree alpha-diversity at an very high level of accuracy of 66% and tree species-richness with 71%. The map with 0.1-degree resolution is a step forward compared to previous maps with one degree resolution and no stratification as to soil and forest. Other, global, mapping exercises also included Amazonia but included far less plot data resulting in a more course-grained pattern, despite the higher map resolution[37,38]. The difference in species richness patterns can be explained in part by our much higher number of plots in Amazonia but also in the way the maps were constructed. Liang et al.[38], for example, used unstratified, interpolated data for soil variables. Such interpolations are invariably dominated by the main forest type in a region (i.e. terra firme in Amazonia). Thus, for fertility the very fertile várzea soils

are downgraded by the infertile terra firme forests in their vicinity, much as Fisher's alpha in those areas was overestimated in our previous map (Supplementary Fig. 5). The map average for that map is very close to that of the main forest types (the three terra firme types, Supplementary Fig. 5c), which have a residual close to zero. All other forest types have negative residuals (Supplementary Fig. 5c), as their Fisher's alpha is generally below that of the regional average. In our new maps, by a-priori assigning plots to their main forest type the predictions are improved and white sand areas and riverine forests clearly emerge as areas with lower diversity and species richness, within richer terra firme (Figs. 1, 2). Overall, the model of Liang et al. explains a much lower percentage of variation in species-richness/ha of our plot data here (28%, Supplementary Fig. 25) and has significant residual variation for forest type (Supplementary Fig. 25b) and region (Supplementary Fig. 25d). In our model residuals for forest type and region are completely lacking (Supplementary Fig. 3b, c, S7b, c), indicating an overall better explanatory power of observed patterns. In addition, other recent maps based on species ranges[39,40] cannot show the detail by soil type, we provide as the modelling is mainly based on smooth, interpolated climatic data. While ref. [40] shows a broadly similar pattern, the map of ref. [39] shows little similarity with the maps provided here.

Forest type has a strong effect on diversity. Diversity is the inverse of species dominance on the plots which was shown to be related to the total area these forest-soil combinations make up in Amazonia[17]. For the seven forest-soil combinations used here, a similar relationship holds (Supplementary Fig. 26). A larger are has a larger species pool and thus higher local richness and diversity[21,41]. We believe that the effect of area is one of the main drivers of the differences in diversity and richness in these forest systems[17,26,42], as also mentioned in the introduction.

Both tree density at plot level and, the interpolated version of it (regional tree density), affected diversity and species-richness, with regional density having a stronger effect. This suggests that density is more than just sampling bias but rather a regional signal where more individuals in a region add up to more species[21], a richer regional species pool and thus higher local richness as well[41]. Both larger area and the higher tree density lead to more regional individuals, leading to a larger regional species pool[21] and thus higher potential local richness[41]. We believe that this is a strong driver of tree alpha-diversity and species-richness of tree plots across Amazonia.

The overall prediction of the maps is very good with low standard errors. For white sand forest and swamp forests, however, the standard error is much higher, leading to less accurate predictions, as was also shown by the leave-out-one test. This appears to be mainly caused by the lower number of plots, as the standard deviation of the mean diversity of terra-firme plots not so much smaller than that of the other forest types (Figs. 1, 2). Thus, the prediction of swamp forest diversity and white sand forest diversity would improve from more plots in these forest types. This is, however, partly caused by how the standard error is calculated—the standard deviation by the square root of the number of items—thus, strongly influenced by the number of plots per forests type. The standard error in Fig. S26, is therefore much related to the number of plots per forest type.

Annual rainfall and cumulative water deficit affected tree species-richness of terra-firme forest, with cumulative water deficit having a slightly stronger effect. The relationship is not strictly linear but more a quantile relationship, where the upper limit of species-richness is determined by the length and strength of the dry season, together driving the cumulative water deficit. This effect was also apparent in a set of 69 0.1-ha plots[6] and in our earlier data[25]. The effect of species filtering by reduced rain

and increased drought was also shown convincingly by studies in western Amazonia[4].

Collecting intensity also had a significant effect on tree species-richness. Still, it added no explained variation to a model with just cumulative water deficit, regional tree density and temperature seasonality. Two areas stand out in collecting intensity: the area surrounding Manaus (Brazil) and French Guiana. These areas also tend to have plots with high species-richness. However, the expected effect does not add to the other variables. It should be noted that even in the final model (Fig. 4) the highest positive residuals are found around Manaus, central Amazonia. Year of collecting had little effect on tree species-richness. We expected richness to increase steadily through time, as more species would have been described, through time[31]. However, as most plots prior to 1980 were primarily located in the poorer eastern Amazonian regions, and the effect was very small, we may suggest that the morpho-species on most of the plots are a sufficient proxy for the actual species richness.

The estimated soil parameters, pH and sum of bases, did not have a particularly strong effect in explaining tree species-richness of terra-firme (see also ref. [43]). This contrasts with earlier findings of a strong relationship between regional soil fertility and composition[5] but supports the results of smaller plots, where actual measured nutrients contributed little to species-richness [6]. The lack of strong relationship between soil fertility and diversity is best visualized by comparing two areas with high diversity (western and central Amazonia) with very contrasting overall fertility[44,45]. We therefore conclude that at large scale water availability is a more important driver for tree species-richness than is soil fertility, see also ref. [4].

The Guyana Shield area, especially the northern part, from Suriname to the upper Rio Negro area, had relatively low species-richness and is the main area with negative residuals for the predictions of the final model (Fig. 4 and all other models, Figs. S17–S25, S27–S29), indicating that this area has lower species-richness than expected by cumulative water deficit, regional tree density and temperature seasonality. Although our regional soil predictors had little predictive power, this is the main area in Amazonia with predominantly sandy, nutrient-poor, soils. Also much of the terra-firme forest here occurs on sandy soils with low clay content that gives them a reddish tinge (called Iwité in the Upper Rio Negro area [W. Magnusson pers comm.]) but having a very different composition than forests on adjacent white sand soils[7,46]. We suggest that the model results would improve with better regional soil maps, especially for this area. Another plausible hypothesis is that the Guianas have been separated geographically from the rest of Amazonia and experienced lower regional species input from the rest of Amazonia during glacial periods, when Amazonia may have consisted of two large forest blocks[47–49] and are still somewhat separated by a "dry transverse belt", the Acarai Mountains (located at the border of the Guianas and Brazil) and the Guyana highlands.

Southern Amazonia and especially the Bolivian forests had modestly low negative residuals in all models that did not include temperature seasonality. The Amazonian forests are thought to have been expanding southward into the Cerrado area only relatively recent see e.g., ref. [50]. Some of these forests may be in the range of only 3000–7000 years old and are still dominated by fast-growing tree species, such as Moraceae and Urticaceae in the Bolivian forests[50,51]. More forests in the transition towards the Cerrado in Brazil, also may still be accumulating species-richness but may face a challenge with increasing drought caused by global warming and droughts. As pre-Columbian inhabitation and forest use and clearing has been perhaps been most prominent in the southern–southwestern border of Amazonia, this may also

have affected diversity negatively. Indeed, in some Bolivian forests, species domesticated by pre-Columbian people, may make up over 60% of all trees[52]. We cannot determine whether the historical changes, the gradient in temperature seasonality, or a combination of the two, are responsible for the lower diversity in that area.

Finally, central Amazonia has a modestly higher richness than all models predicted and is undoubtedly one of the centres of tree species-richness in Amazonia (Figs. 3, 4). It has been proposed that central Amazonia is an area where several biogeographic regions overlap, leading to high richness[53]. Alternatively, a fully random mid-domain effect[54] of overlapping distribution ranges has been suggested for this pattern[25]. However, since most species in Amazonia are rare[55,56] and likely have small ranges[57,58], a random distribution of ranges would lead to a rather flat curve with only an effect at the edges, ruling out a mid-domain effect. The high species richness of central Amazonia is not picked up by other maps[37,38], and this is likely due to the much higher number of plots in our data, leading to a more precise prediction.

Our maps show the spatial distribution of tree diversity and richness of the original forest cover of Amazonia and we have identified drivers for the main patterns. Whereas species-richness may be taken as input for conservation, the composition of Amazonian forests is not homogeneous[4,5,59] and the differences in forest composition have to be taken into account for comprehensive conservation. Forest loss in Amazonia has been increasing since 2014 (http://terrabrasilis.dpi.inpe.br/app/dashboard/deforestation/biomes/amazon/increments). Most deforestation has taken place in the states of Pará, Mato Grosso and Rondônia. These are not the areas on our map with the highest species-richness. However, they may have species that do not occur elsewhere in Amazonia and even with continuous, moderate forest loss, several species may become critically endangered[40,58], and this area may have only very fragmented forest left, which will be vulnerable to drought, fire, hunting and other human impacts[40,58].

In conclusion, seven main forest-soil combinations have a strong effect on tree species-diversity and richness, arguably driven by differences in size and fragmentation of their area and species trade-offs due to very different ecologies. Using location as the only predictor and stratified to account for the four major soil-forest combinations, our spatial model provides the most accurate map of tree diversity in Amazonia to date, explaining close to 70% of variation in tree alpha-diversity and over 70% of the variation in tree species-richness of Amazonian forest plots. Alternatively, a model not using location but with cumulative water deficit, tree density and temperature seasonality explains 43% of the tree species-richness in the terra-firme forest in Amazonia. Over large areas across Amazonia, residuals of this relationship are small and poorly spatially structured, suggesting that much of the residual variation may be local. The poor predictions of the final model in southern Amazonia (notably Bolivian Amazonia) and the northern Guyana Shield may have biogeographic and anthropogenic causes, as in the expanding forests of southern Amazonia and long-time separation in the Guyana Shield area, leading to a complex history and ecology of Amazonian tree species-richness.

## Methods
**Tree data**. Tree-inventory data of undisturbed/old growth forest were taken from the April 2023 version of the Amazon Tree Diversity Network (ATDN) inventory database[55,56]. ATDN stores single inventories for each plot for trees with a diameter at breast height (dbh, 1.30 m) ≥10 cm. A tree is defined as a free-standing woody individual with dbh ≥10 cm[60].

A total of 2220 plots were present in this database, with individuals identified at least at the morpho-species level. We omitted plots smaller than 0.5 ha (138 plots) and larger than 2 ha (36 plots), leaving 2046 plots for all calculations and mapping (Supplementary Fig. 27).

**Modelling density, diversity, and richness patterns**. For each plot, tree density was calculated as the number of stems per ha ($N_{ha}$). Tree species alpha-diversity was expressed as Fisher's alpha, a diversity measure theoretically insensitive to sample size[1], by iteratively solving $\alpha = S/\ln(1 + N/\alpha)$, with N as the total number of individuals and S as the total number of morpho-species per plot. As not all plots represent one hectare, species-richness per ha ($S_{ha}$) was estimated solving for $S_{ha} = \alpha * \ln(1 + N_{ha}/\alpha)$[1]. Note that if a plot is exactly one ha, $S_{ha}$ is exactly equal to S. As both area and the number of individuals (i.e. sample size) have a known positive effect on richness[21], calculating the number of species per ha circumvents the discrepancy in plot size but not differences in density of individuals. Plots with higher densities will still have, on average, higher richness. To account for the latter, we also calculated the number of species in (a random sample of) 500 individuals from each plot as $S_{500} = \alpha * \ln(1 + 500/\alpha)$[1].

Finally, the spatial predictions of tree density (number of trees/ha), tree alpha-diversity and tree species-richness for the Amazon lowland forest were plotted on a map with a resolution of 0.1 degree (11 × 11 km, Supplementary Fig. 28a)[58,60], based on the original forest extent of Amazonia, stratified into the major soils corresponding to the major forest-soil combinations used[28,55] (Supplementary Fig. 28b).

For this, we constructed a simplified soil map based on the Soil and Terrain database for Latin America and the Caribbean[27,28,55], to match this division. We aggregated all soil types into 1) poor white sand areas using FAO soil types Podzols (PZ) and Albic arenosols (ARa); 2) floodplains (várzea, igapó) using Gleysoils (Gl), Fluvisols (Fl); 3) swamps, using all Histosols (Hs); 4) and the remainder as soils supporting terra-firme forest. The ATDN plots were subdivided following this soil-flooding-based approach according to the four categories, that do justice to the major soil-forest combinations, while ensuring sufficient plots for interpolation by category: 1) non-flooded Terra-firme (1443 plots used), 2) floodplain forests (várzea [241] and igapó [222]), 3) very nutrient poor white sand podzols (95), and 4) permanently inundated/waterlogged swamps (46) (Supplementary Fig. 28).

For our spatial interpolations we used loess regression, using only longitude, latitude and their interaction as independent variables and tree density, tree alpha-diversity and species-richness as the dependent variables. For all loess regressions we used a span of 0.2, a 2nd degree polynomial, and no extrapolation. Kriging was not possible as at several locations with multiple plots most variation was already locally present, so the semivariograms showed no range.

For each of the four categories (terra-firme, várzea plus igapó, podzols, swamps), we constructed a separate spatial interpolation model of tree density, tree alpha-diversity and tree species-richness across Amazonia.

For example, for tree alpha-diversity, we made a single spatial interpolation for all plots located on white-sand podzols. This interpolation was then used to predict the value for tree alpha-diversity for each grid cell on the soil map considered to be white-sand podzol (Supplementary Fig. 28b, yellow pixels for white-sand). The same was done for all seasonally flooded forest plots (várzea + igapó, combined to have sufficient plots), all swamp plots, and all plots established on terra-firme. Whereas the soil grid[27] is based on the major hydrology/soil type; the soil type of

the plots was determined independently of this map and based on field observations of those who established the plot. Consequently, it is possible that a plot classified by observers as white-sand podzols is located in a grid cell classified as terra-firme on the map. Regardless, it was used in the white-sand spatial model as the field observations are considered to be correct. For a visual explanation of this method, see Supplementary Fig. 29. As we allowed no extrapolation, pixels too far from the plots were not given a value. As a 2nd degree polynomial may produce upward and downward exaggerations, values higher than the observed maximum in the data were set to the maximum value and those lower than the minimum to the minimum value.

**Testing the model fit**. We calculated the percentage of variation as explained by the combination of the spatial models for each variable (tree density, tree alpha-diversity and tree species-richness), by analysing the observed and predicted values together, using a simple linear regression. We tested the goodness of prediction by mapping the standard error of the loess regression, also examining them by region and forest type. We tested for autocorrelation in the residuals, using the function Moran.I(), in the *ape* package distribution[61] to further assess the validity of the model predictions[62] and mapped the residuals to asses potential residual spatial signal. A histogram was constructed of all values for each variable, as well as boxplot by region and forest type. A final test was performed by producing 2046 maps for species richness/ha, where each plot was omitted in one run (a leave-one-out procedure). This map was then used to predict the species-richness/ha for the plot that was left out and can be considered a non-biased estimate of the quality of the resulting map. We modelled the effect of climate and large-scale patterns of soil nutrient richness but for terra-firme only, as this forest type had the highest number of plots.

To assess the effect of the number of individuals on tree alpha-diversity and tree species-richness of terra-firme forest, we used the local tree density (i.e. the number of trees/ha for each plot) and the interpolated stem density (also expressed as trees/ha), which is a measure of the average density in an area surrounding the plots. We assume that large areas with higher density, having more individuals, have a higher species pool[21], resulting in higher species-richness at the plot level. We use the term regional tree density for this.

Climatic data was extracted by plot location from the grid data from Worldclim 2[63]. The cumulative water deficit was calculated following Chave et al.[64] and can be considered a parameter of the strength of the dry season. Soil fertility (log[sum of bases]) was extracted from the latest Amazonia-wide soil-fertility map[65]. We used sum of bases rather than the often-used CEC (Cation Exchange Capacity), as the latter includes the full exchange complex, which on acid tropical soils often includes a large portion of $Al^{3+}$ and $H^+$. Soil acidity (pH) is also an often-used index of soil fertility (a low pH being infertile). We extracted pH data from Soterlac[27], ISRIC wise[66], RAINFOR sites[44], and refs. [67–69]. For the sum of bases and pH, we created a loess interpolation model, based on all data available (data availability differed between sum of bases and pH), as described above (Supplementary Fig. 30). We then estimated the sum of bases and pH for each plot based on the loess interpolation.

Collecting intensity was based on the 530,025 unique herbarium collections of Amazonian trees from ref. [31], using the standard Kernel density function of R with Gaussian smoothing and adjustment of 0.2[70] (Supplementary Fig. 31A). The latter is comparable with the loess span = 0.2 used in our loess interpolation (Supplementary Fig. 1B). The year of the

establishment was known for most plots. If unknown, we used the year of publication minus one year.

We analysed the effect of abiotic variables on tree species-richness only, for two reasons: 1) Species-richness is much easier interpretable than Fisher's alpha, and 2) it has a very strong relationship with Fisher's alpha. We also used quantile regression[71,72], as 1) quantile regression is much less sensitive to outliers, with quantile regression using tau = 0.5 being identical to least absolute deviation regression (i.e. line dividing the data at 50% minimizes, as follows from the name, the absolute deviation from that line) and 2) because it allows flexibility of using other quantiles as well. We used tau = 0.9, which produces the line with 90% of the data below and 10% above it, minimizing the absolute deviation. This line can be seen as the maximum the dependent variable can achieve for a value of the independent variable and has been used successfully to demonstrate the effect of dry season length of the Amazonian forest before[25].

All analyses were carried out in the R programming environment[70], mostly with custom made scripts, using the libraries *ape*[61], *jpeg*[73], *raster*[74], *rgdal*[75], *quantreg*[76], and *vegan*[77].

**Statistics and reproducibility**. All tests were carried out with all plots ($n = 2047$) or all terra firme plots (1441). All tests and data are available in the online supplementary material (see below) and can thus be reproduced.

**Reporting summary**. Further information on research design is available in the Nature Portfolio Reporting Summary linked to this article.

## Data availability

All plot metadata, tree density, diversity and richness data by plot (to make all figures and supplementary figures) and raster maps of tree density, tree alpha-diversity and tree species-richness, two large images of the tree-diversity and richness map are publicly available through FigShare[78]. The code to make all figures and supplementary figures is also available on FigShare[78]. Additional data is available upon reasonable request by contacting the corresponding author.

## Code availability

All custom R code used in the analysis and visualization of the data is publicly available through FigShare[78]. This code can be used to produce all figures and supplementary figures.

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

## Acknowledgements

This paper is the result of the work of hundreds of different scientists and research institutions in the Amazon over the past 80 years. Without their hard work this analysis would have been impossible. We thank Charles Zartman for the use of plots from Jutai. H.t.S., V.F.G. and R.S. were supported by grant 407232/2013-3 - PVE - MEC/MCTI/CAPES/CNPq/FAPs; P.I.P. had support for this work from CNPq (productivity grant 310885/2017-5) and FAPESP (research grant #09/53413-5); C.B. was supported by grant FAPESP 95/3058-0 - CRS 068/96 WWF Brasil - The Body Shop; D.S., J.F.M., J.E., P.P. and J.C. benefited from an "Investissement d'Avenir" grant managed by the Agence Nationale de la Recherche (CEBA: ANR-10-LABX-25-01); H.L.Q./M.A.P./J.L.L.M. received financial supported by MCT/CNPq/CT-INFRA/GEOMA #550373/2010-1 and # 457515/2012-0, and J.L.L.M. were supported by grant CAPES/PDSE # 88881.135761/2016-01 and CAPES/Fapespa #1530801; The Brazilian National Research Council (CNPq) provided a productivity grant to E.M.V. (Grant 308040/2017-1); Floristic identification in plots in the RAINFOR forest monitoring network has been supported by the Natural Environment Research Council (grants NE/B503384/1, NE/D01025X/1, NE/I02982X/1, NE/F005806/1, NE/D005590/1 and NE/I028122/1) and the Gordon and Betty Moore Foundation; B.M.F. is funded by FAPESP grant 2016/25086-3. B.S.M., B.H.M.J. and O.L.P. were supported by grants CNPq/CAPES/FAPS/BC-Newton Fund #441244/2016-5 and FAPEMAT/0589267/2016; T.W.H. was funded by National Science Foundation grant DEB-1556338. B.G.L. was supported by FAPESP, grant #2015/24554-0, and #2019/03379-4. W.E.M.: Plots in the PPBio system were financed by the INCT for Amazonian Biodiversity (CENBAM); the Program for Biodiversity Research in Western Amazonia (PPBio-AmOc) and a Productivity Grant (PQ - 301873/2016-0). Finally, we thank four reviewers for their questions and comments, which have greatly improved the manuscript.

## Author contributions

H.t.S. and N.C.A.P. conceived the study. H.t.S. carried out the analyses and wrote the first manuscript version. All authors contributed essential plot data and commented on the manuscript and gave permission for its publication.

## Competing interests

The authors declare no competing interests.

## Additional information

Hans ter Steege [1,2✉], Nigel C. A. Pitman[3], Iêda Leão do Amaral[4], Luiz de Souza Coelho[4], Francisca Dionízia de Almeida Matos[4], Diógenes de Andrade Lima Filho[4], Rafael P. Salomão[5,6], Florian Wittmann[7,8], Carolina V. Castilho[9], Juan Ernesto Guevara[10,11], Marcelo de Jesus Veiga Carim[12], Oliver L. Phillips[13], William E. Magnusson[14], Daniel Sabatier[15], Juan David Cardenas Revilla[4], Jean-François Molino[15], Mariana Victória Irume[4], Maria Pires Martins[4], José Renan da Silva Guimarães[16], José Ferreira Ramos[4], Olaf S. Bánki[1], Maria Teresa Fernandez Piedade[8], Dairon Cárdenas López[17,153], Domingos de Jesus Rodrigues[18], Layon O. Demarchi[8], Jochen Schöngart[8], Everton José Almeida[19], Luciane Ferreira Barbosa[19], Larissa Cavalheiro[19], Márcia Cléia Vilela dos Santos[19], Bruno Garcia Luize[20], Evlyn Márcia Moraes de Leão Novo[21], Percy Núñez Vargas[22], Thiago Sanna Freire Silva[23], Eduardo Martins Venticinque[24], Angelo Gilberto Manzatto[25], Neidiane Farias Costa Reis[26], John Terborgh[27,28], Katia Regina Casula[26], Euridice N. Honorio Coronado[29,30], Abel Monteagudo Mendoza[22,31], Juan Carlos Montero[4,32], Flávia R. C. Costa[14], Ted R. Feldpausch[13,33], Adriano Costa Quaresma[7,8], Nicolás Castaño Arboleda[17], Charles Eugene Zartman[4], Timothy J. Killeen[34], Beatriz S. Marimon[35], Ben Hur Marimon-Junior[35], Rodolfo Vasquez[31], Bonifacio Mostacedo[36], Rafael L. Assis[37], Chris Baraloto[38], Dário Dantas do Amaral[6], Julien Engel[15,38], Pascal Petronelli[39], Hernán Castellanos[40], Marcelo Brilhante de Medeiros[41], Marcelo Fragomeni Simon[41], Ana Andrade[42], José Luís Camargo[42], William F. Laurance[28], Susan G. W. Laurance[28], Lorena Maniguaje Rincón[4], Juliana Schietti[4], Thaiane R. Sousa[43], Emanuelle de Sousa Farias[44,45], Maria Aparecida Lopes[46], José Leonardo Lima Magalhães[47,48],

Henrique Eduardo Mendonça Nascimento[4], Helder Lima de Queiroz[49], Gerardo A. Aymard C.[50], Roel Brienen[13], Pablo R. Stevenson[51], Alejandro Araujo-Murakami[52], Tim R. Baker[13], Bruno Barçante Ladvocat Cintra[53], Yuri Oliveira Feitosa[54], Hugo F. Mogollón[55], Joost F. Duivenvoorden[56], Carlos A. Peres[57], Miles R. Silman[58], Leandro Valle Ferreira[6], José Rafael Lozada[59], James A. Comiskey[60,61], Freddie C. Draper[62], José Julio de Toledo[63], Gabriel Damasco[64], Roosevelt García-Villacorta[65,66], Aline Lopes[67], Alberto Vicentini[14], Fernando Cornejo Valverde[68], Alfonso Alonso[61], Luzmila Arroyo[52], Francisco Dallmeier[61], Vitor H. F. Gomes[69,70], Eliana M. Jimenez[71], David Neill[72], Maria Cristina Peñuela Mora[73], Janaína Costa Noronha[18], Daniel P. P. de Aguiar[74,75], Flávia Rodrigues Barbosa[18], Yennie K. Bredin[76,77], Rainiellen de Sá Carpanedo[18], Fernanda Antunes Carvalho[14,78], Fernanda Coelho de Souza[13,14], Kenneth J. Feeley[79,80], Rogerio Gribel[4], Torbjørn Haugaasen[76], Joseph E. Hawes[81], Marcelo Petratti Pansonato[4,82], Marcos Ríos Paredes[83], Jos Barlow[84], Erika Berenguer[84,85], Izaias Brasil da Silva[86], Maria Julia Ferreira[87], Joice Ferreira[48], Paul V. A. Fine[88], Marcelino Carneiro Guedes[89], Carolina Levis[90], Juan Carlos Licona[32], Boris Eduardo Villa Zegarra[91], Vincent Antoine Vos[92], Carlos Cerón[93], Flávia Machado Durgante[7,8], Émile Fonty[15,94], Terry W. Henkel[95], John Ethan Householder[7], Isau Huamantupa-Chuquimaco[96], Edwin Pos[2,97], Marcos Silveira[98], Juliana Stropp[99], Raquel Thomas[100], Doug Daly[101], Kyle G. Dexter[102,103], William Milliken[104], Guido Pardo Molina[92], Toby Pennington[33,103], Ima Célia Guimarães Vieira[6], Bianca Weiss Albuquerque[8], Wegliane Campelo[63], Alfredo Fuentes[105,106], Bente Klitgaard[107], José Luis Marcelo Pena[108], J. Sebastián Tello[106], Corine Vriesendorp[3], Jerome Chave[109], Anthony Di Fiore[110,111], Renato Richard Hilário[63], Luciana de Oliveira Pereira[33], Juan Fernando Phillips[112], Gonzalo Rivas-Torres[111,113], Tinde R. van Andel[1,114], Patricio von Hildebrand[115], William Balee[116], Edelcilio Marques Barbosa[4], Luiz Carlos de Matos Bonates[4], Hilda Paulette Dávila Doza[83], Ricardo Zárate Gómez[117], Therany Gonzales[118], George Pepe Gallardo Gonzales[83], Bruce Hoffman[119], André Braga Junqueira[120], Yadvinder Malhi[121], Ires Paula de Andrade Miranda[4], Linder Felipe Mozombite Pinto[83], Adriana Prieto[122], Agustín Rudas[122], Ademir R. Ruschel[48], Natalino Silva[123], César I. A. Vela[124], Egleé L. Zent[125], Stanford Zent[125], Angela Cano[51,126], Yrma Andreina Carrero Márquez[127], Diego F. Correa[51,128], Janaina Barbosa Pedrosa Costa[89], Bernardo Monteiro Flores[90], David Galbraith[13], Milena Holmgren[129], Michelle Kalamandeen[130], Guilherme Lobo[131], Luis Torres Montenegro[132], Marcelo Trindade Nascimento[133], Alexandre A. Oliveira[82], Maihyra Marina Pombo[4], Hirma Ramirez-Angulo[134], Maira Rocha[8], Veridiana Vizoni Scudeller[135], Rodrigo Sierra[136], Milton Tirado[136], Maria Natalia Umaña[137], Geertje van der Heijden[138], Emilio Vilanova Torre[134,139], Manuel Augusto Ahuite Reategui[140], Cláudia Baider[82,141], Henrik Balslev[142], Sasha Cárdenas[51], Luisa Fernanda Casas[51], María José Endara[10], William Farfan-Rios[22,58], Cid Ferreira[4,153], Reynaldo Linares-Palomino[61], Casimiro Mendoza[143,144], Italo Mesones[88], Germaine Alexander Parada[52], Armando Torres-Lezama[134], Ligia Estela Urrego Giraldo[145], Daniel Villarroel[52,146], Roderick Zagt[147], Miguel N. Alexiades[148], Edmar Almeida de Oliveira[35], Karina Garcia-Cabrera[58], Lionel Hernandez[40], Walter Palacios Cuenca[149], Susamar Pansini[26], Daniela Pauletto[150], Freddy Ramirez Arevalo[151], Adeilza Felipe Sampaio[26], Elvis H. Valderrama Sandoval[151,152], Luis Valenzuela Gamarra[31], Aurora Levesley[13], Georgia Pickavance[13] & Karina Melgaço[13]

[1]Naturalis Biodiversity Center, PO Box 9517, Leiden 2300 RA, The Netherlands. [2]Quantitative Biodiversity Dynamics, Utrecht University, Padualaan 8, Utrecht 3584 CH, The Netherlands. [3]Science and Education, The Field Museum, 1400 S. Lake Shore Drive, Chicago, IL 60605-2496, USA. [4]Coordenação de Biodiversidade, Instituto Nacional de Pesquisas da Amazônia - INPA, Av. André Araújo, 2936, Petrópolis, Manaus, AM 69067-375, Brazil. [5]Programa Professor Visitante Nacional Sênior na Amazônia - CAPES, Universidade Federal Rural da Amazônia, Av. Perimetral, s/n, Belém, PA, Brazil. [6]Coordenação de Botânica, Museu Paraense Emílio Goeldi, Av. Magalhães Barata 376, C.P. 399, Belém, PA 66040, Brazil. [7]Wetland Department, Institute of Geography and Geoecology, Karlsruhe Institute of Technology - KIT, Josefstr.1, Rastatt D-76437, Germany. [8]Ecology, Monitoring and Sustainable Use of Wetlands (MAUA), Instituto Nacional de Pesquisas da Amazônia - INPA, Av. André Araújo, 2936, Petrópolis, Manaus, AM 69067-375, Brazil. [9]Centro de Pesquisa Agroflorestal de Roraima, Embrapa Roraima, BR 174, km 8 – Distrito Industrial, Boa Vista, RR 69301-970, Brazil. [10]Grupo de Investigación en Biodiversidad, Medio Ambiente y Salud-BIOMAS, Universidad de las Américas,

Campus Queri, Quito, Ecuador. [11]Keller Science Action Center, The Field Museum, 1400 S. Lake Shore Drive, Chicago, IL 60605-2496, USA. [12]Departamento de Botânica, Instituto de Pesquisas Científicas e Tecnológicas do Amapá - IEPA, Rodovia JK, Km 10, Campus do IEPA da Fazendinha, Macapá, AP 68901-025, Brazil. [13]School of Geography, University of Leeds, Woodhouse Lane, Leeds LS2 9JT, UK. [14]Coordenação de Pesquisas em Ecologia, Instituto Nacional de Pesquisas da Amazônia - INPA, Av. André Araújo, 2936, Petrópolis, Manaus, AM 69067-375, Brazil. [15]AMAP, IRD, Cirad, CNRS, INRAE, Université de Montpellier, Montpellier F-34398, France. [16]Amcel Amapá Florestal e Celulose S.A, Rua Claudio Lucio - S/N, Novo Horizonte, Santana, AP 68927, Brazil. [17]Herbario Amazónico Colombiano, Instituto SINCHI, Calle 20 No 5-44, Bogotá, DC, Colombia. [18]ICNHS, Federal University of Mato Grosso, Av. Alexandre Ferronato 1200, Setor Industrial, Sinop, MT 78.557-267, Brazil. [19]ICNHS, Universidade Federal de Mato Grosso, Av. Alexandre Ferronato, 1200, Sinop, MT 78557-267, Brazil. [20]Departamento de Biologia Vegetal, Instituto de Biologia, Universidade Estadual de Campinas – UNICAMP, CP 6109, Campinas, SP 13083-970, Brazil. [21]Divisao de Sensoriamento Remoto – DSR, Instituto Nacional de Pesquisas Espaciais – INPE, Av. dos Astronautas, 1758, Jardim da Granja, São José dos Campos, SP 12227-010, Brazil. [22]Herbario Vargas, Universidad Nacional de San Antonio Abad del Cusco, Avenida de la Cultura, Nro 733, Cusco, Cuzco, Peru. [23]Biological and Environmental Sciences, University of Stirling, Stirling FK9 4LA, UK. [24]Centro de Biociências, Departamento de Ecologia, Universidade Federal do Rio Grande do Norte, Av. Senador Salgado Filho, 3000, Natal, RN 59072-970, Brazil. [25]Departamento de Biologia, Universidade Federal de Rondônia, Rodovia BR 364 s/n Km 9,5 - Sentido Acre, Unir, Porto Velho, RO 76.824-027, Brazil. [26]Programa de Pós- Graduação em Biodiversidade e Biotecnologia PPG- Bionorte, Universidade Federal de Rondônia, Campus Porto Velho Km 9,5 bairro Rural, Porto Velho, RO 76.824-027, Brazil. [27]Department of Biology and Florida Museum of Natural History, University of Florida, Gainesville, FL 32611, USA. [28]Centre for Tropical Environmental and Sustainability Science and College of Science and Engineering, James Cook University, Cairns, Queensland 4870, Australia. [29]Instituto de Investigaciones de la Amazonía Peruana (IIAP), Av. A. Quiñones km 2,5, Iquitos, Loreto 784, Peru. [30]School of Geography and Sustainable Development, University of St Andrews, Irvine Building, St Andrews KY16 9AL, UK. [31]Jardín Botánico de Missouri, Oxapampa, Pasco, Peru. [32]Instituto Boliviano de Investigacion Forestal, Av. 6 de agosto #28, Km. 14, Doble via La Guardia, Casilla, Santa Cruz, 6204 Santa Cruz, Bolivia. [33]Geography, College of Life and Environmental Sciences, University of Exeter, Rennes Drive, Exeter EX4 4RJ, UK. [34]Agteca-Amazonica, Santa Cruz, Bolivia. [35]Programa de Pós-Graduação em Ecologia e Conservação, Universidade do Estado de Mato Grosso, Nova Xavantina, MT, Brazil. [36]Facultad de Ciencias Agrícolas, Universidad Autónoma Gabriel René Moreno, Santa Cruz, Santa Cruz, Bolivia. [37]Biodiversity and Ecosystem Services, Instituto Tecnológico Vale, Belém, Pará, Brazil. [38]International Center for Tropical Botany (ICTB) Department of Biological Sciences, Florida International University, 11200 SW 8th Street, OE 243, Miami, FL 33199, USA. [39]Cirad UMR Ecofog, AgrosParisTech, CNRS, INRAE, Univ Guyane, Campus agronomique, Kourou Cedex 97379, France. [40]Centro de Investigaciones Ecológicas de Guayana, Universidad Nacional Experimental de Guayana, Calle Chile, urbaniz Chilemex, Puerto Ordaz, Bolivar, Venezuela. [41]Embrapa Recursos Genéticos e Biotecnologia, Parque Estação Biológica, Prédio da Botânica e Ecologia, Brasilia, DF 70770-917, Brazil. [42]Projeto Dinâmica Biológica de Fragmentos Florestais, Instituto Nacional de Pesquisas da Amazônia - INPA, Av. André Araújo, 2936, Petrópolis, Manaus, AM 69067-375, Brazil. [43]Programa de Pós-Graduação em Ecologia, Instituto Nacional de Pesquisas da Amazônia - INPA, Av. André Araújo, 2936, Petrópolis, Manaus, AM 69067-375, Brazil. [44]Laboratório de Ecologia de Doenças Transmissíveis da Amazônia (EDTA), Instituto Leônidas e Maria Deane, Fiocruz, Rua Terezina, 476, Adrianópolis, Manaus, AM 69060-001, Brazil. [45]Programa de Pós-graduação em Biodiversidade e Saúde, Instituto Oswaldo Cruz - IOC/FIOCRUZ, Pav. Arthur Neiva – Térreo, Av. Brasil, 4365 – Manguinhos, Rio de Janeiro, RJ 21040-360, Brazil. [46]Instituto de Ciências Biológicas, Universidade Federal do Pará, Av. Augusto Corrêa 01, Belém, PA 66075, Brazil. [47]Programa de Pós-Graduação em Ecologia, Universidade Federal do Pará, Av. Augusto Corrêa 01, Belém, PA 66075-110, Brazil. [48]Empresa Brasileira de Pesquisa Agropecuária, Embrapa Amazônia Oriental, Trav. Dr. Enéas Pinheiro s/n°, Belém, PA 66095-903, Brazil. [49]Diretoria Técnico-Científica, Instituto de Desenvolvimento Sustentável Mamirauá, Estrada do Bexiga, 2584, Tefé, AM 69470-000, Brazil. [50]Programa de Ciencias del Agro y el Mar, Herbario Universitario (PORT), UNELLEZ-Guanare, Guanare, Portuguesa 3350, Venezuela. [51]Laboratorio de Ecología de Bosques Tropicales y Primatología, Universidad de los Andes, Carrera 1 # 18a-10, Bogotá, DC 111711, Colombia. [52]Museo de Historia Natural Noel Kempff Mercado, Universidad Autónoma Gabriel Rene Moreno, Avenida Irala 565 Casilla Post al 2489, Santa Cruz, Santa Cruz, Bolivia. [53]Birmingham Institute for Forest Research, University of Birmingham, Edgbaston, Birmingham B15 2TT, UK. [54]Programa de Pós-Graduação em Biologia (Botânica), Instituto Nacional de Pesquisas da Amazônia - INPA, Av. André Araújo, 2936, Petrópolis, Manaus 69067, Brazil. [55]Endangered Species Coalition, 8530 Geren Rd., Silver Spring, MD 20901, USA. [56]Institute of Biodiversity and Ecosystem Dynamics, University of Amsterdam, Sciencepark 904, Amsterdam 1098 XH, The Netherlands. [57]School of Environmental Sciences, University of East Anglia, Norwich NR4 7TJ, UK. [58]Biology Department and Center for Energy, Environment and Sustainability, Wake Forest University, 1834 Wake Forest Rd, Winston Salem, NC 27106, USA. [59]Facultad de Ciencias Forestales y Ambientales, Instituto de Investigaciones para el Desarrollo Forestal, Universidad de los Andes, Via Chorros de Milla, 5101, Mérida, Mérida, Venezuela. [60]Inventory and Monitoring Program, National Park Service, 120 Chatham Lane, Fredericksburg, VA 22405, USA. [61]Center for Conservation and Sustainability, Smithsonian Conservation Biology Institute, 1100 Jefferson Dr. SW, Suite 3123, Washington, DC 20560-0705, USA. [62]Department of Geography and Planning, University of Liverpool, Liverpool L69 3BX, UK. [63]Universidade Federal do Amapá, Ciências Ambientais, Rod. Juscelino Kubitschek km2, Macapá, AP 68902-280, Brazil. [64]Gothenburg Global Biodiversity Centre, University of Gothenburg, Carl Skottbergs gata 22b, Gothenburg 413 19, Sweden. [65]Programa Restauración de Ecosistemas (PRE), Centro de Innovación Científica Amazónica (CINCIA), Jr. Cajamarca Cdra. 1 s/n, Tambopata, Madre de Dios, Peru. [66]Peruvian Center for Biodiversity and Conservation (PCBC), Iquitos, Loreto, Peru. [67]Department of Ecology, Institute of Biological Sciences, University of Brasilia, Brasilia, DF 70904-970, Brazil. [68]Andes to Amazon Biodiversity Program, Madre de Dios, Madre de Dios, Peru. [69]Escola de Negócios Tecnologia e Inovação, Centro Universitário do Pará, Belém, PA, Brazil. [70]Environmental Science Program, Geosciences Department, Universidade Federal do Pará, Rua Augusto Corrêa 01, Belém, PA 66075-110, Brazil. [71]Grupo de Ecología y Conservación de Fauna y Flora Silvestre, Instituto Amazónico de Investigaciones Imani, Universidad Nacional de Colombia sede Amazonia, Leticia, Amazonas, Colombia. [72]Universidad Estatal Amazónica, Puyo, Pastaza, Ecuador. [73]Universidad Regional Amazónica IKIAM, Km 7 via Muyuna, Tena, Napo, Ecuador. [74]Procuradoria-Geral de Justiça, Ministério Público do Estado do Amazonas, Av. Coronel Teixeira, 7995, Manaus, AM 69037-473, Brazil. [75]Coordenação de Dinâmica Ambiental, Instituto Nacional de Pesquisas da Amazônia - INPA, Av. André Araújo, 2936, Petrópolis, Manaus, AM 69067-375, Brazil. [76]Norwegian University of Life Sciences (NMBU), Faculty of Environmental Sciences and Natural Resource Management, P.O. Box 5003 NMBU, Aas, 1432 Aas, Norway. [77]Norwegian Institute for Nature Research (NINA), Sognsveien 68, Oslo, 0855 Oslo, Norway. [78]Universidade Federal de Minas Gerais, Instituto de Ciências Biológicas, Departamento de Genética, Ecologia e Evolução, Av. Antônio Carlos, 6627 Pampulha, Belo Horizonte, MG 31270-901, Brazil. [79]Department of Biology, University of Miami, Coral Gables, FL 33146, USA. [80]Fairchild Tropical Botanic Garden, Coral Gables, FL 33156, USA. [81]Institute of Science and Environment, University of Cumbria, Ambleside, Cumbria LA22 9BB, UK. [82]Instituto de Biociências - Dept. Ecologia, Universidade de Sao Paulo - USP, Rua do Matão, Trav. 14, no. 321, Cidade Universitária, São Paulo, SP 05508-090, Brazil. [83]Servicios de Biodiversidad EIRL, Jr. Independencia 405, Iquitos, Loreto 784, Peru. [84]Lancaster Environment Centre, Lancaster University, Lancaster, Lancashire LA1 4YQ, UK. [85]Environmental Change Institute, University of Oxford, Oxford, Oxfordshire OX1 3QY, UK. [86]Postgraduate program in Biodiversity and Biotechnology – Bionorte, Federal University of Acre, Rodovia 364, km 4.5, Distrito industrial, Rio Branco, AC 69900-000, Brazil. [87]Postgraduate program in Ethnobiology and Nature Conservation, Federal Rural University of

**13**

Pernambuco (UFRPE), Rua Dom Manuel de Medeiros, s/n, Dois Irmãos, Pernambuco, PB 52171-900, Brazil. [88]Department of Integrative Biology, University of California, Berkeley, CA 94720-3140, USA. [89]Empresa Brasileira de Pesquisa Agropecuária, Embrapa Amapá, Rod. Juscelino Kubitschek km 5, Macapá, AP 68903-419, Brazil. [90]Graduate Program in Ecology, Federal University of Santa Catarina (UFSC), Campus Universitário - Córrego Grande, Florianópolis, SC 88040-900, Brazil. [91]Direccíon de Evaluación Forestal y de Fauna Silvestre, Av. Javier Praod Oeste 693, Magdalena del Mar, Peru. [92]Instituto de Investigaciones Forestales de la Amazonía, Universidad Autónoma del Beni José Ballivián, Campus Universitario Final, Av. Ejercito, Riberalta, Beni, Bolivia. [93]Escuela de Biología Herbario Alfredo Paredes, Universidad Central, Ap. Postal, 17.01.2177 Quito, Pichincha, Ecuador. [94]Direction régionale de la Guyane, Office national des forêts, Cayenne 97300, French Guiana. [95]Department of Biological Sciences, California State Polytechnic University, 1 Harpst Street, Arcata, CA 95521, USA. [96]Herbario HAG, Universidad Nacional Amazónica de Madre de Dios (UNAMAD), Av. Jorge Chávez, 1160 Puerto Maldonado, Madre de Dios, Peru. [97]Utrecht University Botanic Gardens, P.O. Box 80162, Utrecht 3508 TD, The Netherlands. [98]Centro de Ciências Biológicas e da Natureza, Universidade Federal do Acre, Rodovia BR 364, Km 4, s/n, Distrito Industrial, Rio Branco, AC 69915-559, Brazil. [99]Museo Nacional de Ciencias Naturales (MNCN-CSIC), C. de José Gutiérrez Abascal 2, Madrid 28006, Spain. [100]Iwokrama International Centre for Rain Forest Conservation and Development, Georgetown, Guyana. [101]New York Botanical Garden, 2900 Southern Blvd, Bronx, New York, NY 10458-5126, USA. [102]School of Geosciences, University of Edinburgh, 201 Crew Building, King's Buildings, Edinburgh EH9 3JN, UK. [103]Tropical Diversity Section, Royal Botanic Garden Edinburgh, 20a Inverleith Row, Edinburgh, Scotland EH3 5LR, UK. [104]Department for Ecosystem Stewardship, Royal Botanic Gardens, Kew, Richmond, Surrey TW9 3AE, UK. [105]Herbario Nacional de Bolivia, Universitario UMSA, Casilla 10077 Correo Central, La Paz, La Paz, Bolivia. [106]Center for Conservation and Sustainable Development, Missouri Botanical Garden, P.O. Box 299, St. Louis, MO 63166-0299, USA. [107]Department for Accelerated Taxonomy, Royal Botanic Gardens, Kew, Richmond, Surrey TW9 3AE, UK. [108]Universidad Nacional de Jaén, Carretera Jaén San Ignacio Km 23, Jaén, Cajamarca 06801, Peru. [109]Laboratoire Evolution et Diversité Biologique, CNRS and Université Paul Sabatier, UMR 5174 EDB, Toulouse 31000, France. [110]Department of Anthropology, University of Texas at Austin, SAC 5.150, 2201 Speedway Stop C3200, Austin, TX 78712, USA. [111]Estación de Biodiversidad Tiputini, Colegio de Ciencias Biológicas y Ambientales, Universidad San Francisco de Quito-USFQ, Quito, Pichincha, Ecuador. [112]Fundación Puerto Rastrojo, Cra 10 No. 24-76 Oficina, 1201 Bogotá, DC, Colombia. [113]Department of Wildlife Ecology and Conservation, University of Florida, 110 Newins-Ziegler Hall, Gainesville, FL 32611, USA. [114]Biosystematics group, Wageningen University, Droevendaalsesteeg 1, Wageningen 6708 PB, The Netherlands. [115]Fundación Estación de Biología, Cra 10 No. 24-76 Oficina, 1201 Bogotá, DC, Colombia. [116]Department of Anthropology, Tulane University, 101 Dinwiddie Hall, 6823 St. Charles Avenue, New Orleans, LA 70118, USA. [117]PROTERRA, Instituto de Investigaciones de la Amazonía Peruana (IIAP), Av. A. Quiñones km 2,5, Iquitos, Loreto 784, Peru. [118]ACEER Foundation, Jirón Cusco N° 370, Puerto Maldonado, Madre de Dios, Peru. [119]Amazon Conservation Team, 4211 North Fairfax Drive, Arlington, VA 22203, USA. [120]Institut de Ciència i Tecnologia Ambientals, Universitat Autònoma de Barcelona, Bellaterra, Barcelona 08193, Spain. [121]Environmental Change Institute, Oxford University Centre for the Environment, Dyson Perrins Building, South Parks Road, Oxford, England OX1 3QY, UK. [122]Instituto de Ciencias Naturales, Universidad Nacional de Colombia, Apartado, 7945 Bogotá, DC, Colombia. [123]Instituto de Ciência Agrárias, Universidade Federal Rural da Amazônia, Av. Presidente Tancredo Neves 2501, Belém, PA 66.077-830, Brazil. [124]Escuela Profesional de Ingeniería Forestal, Universidad Nacional de San Antonio Abad del Cusco, Jirón San Martín 451, Puerto Maldonado, Madre de Dios, Peru. [125]Laboratory of Human Ecology, Instituto Venezolano de Investigaciones Científicas - IVIC, Ado 20632, Caracas, DC 1020A, Venezuela. [126]Cambridge University Botanic Garden, Cambridge University, 1 Brookside, Cambridge CB2 1JE, UK. [127]Programa de Maestria de Manejo de Bosques, Universidad de los Andes, Via Chorros de Milla, 5101, Mérida, Venezuela. [128]Centre for Biodiversity and Conservation Science CBCS, The University of Queensland, Brisbane, QLD 4072, Australia. [129]Resource Ecology Group, Wageningen University & Research, Droevendaalsesteeg 3a, Lumen, building number 100, Wageningen, Gelderland 6708 PB, The Netherlands. [130]School of Earth, Environment and Society, McMaster University, 1280 Main Street West, Hamilton, Ontario L8S 4K1, Canada. [131]Núcleo de Estudos e Pesquisas Ambientais, Universidade Estadual de Campinas – UNICAMP, CP 6109, Campinas, SP 13083-867, Brazil. [132]Herbarium Amazonense (AMAZ), Universidad Nacional de la Amazonia Peruana, Calle Pebas/Nanay, Iquitos, Loreto, Peru. [133]Laboratório de Ciências Ambientais, Universidade Estadual do Norte Fluminense, Av. Alberto Lamego 2000, Campos dos Goytacazes, RJ 28013-620, Brazil. [134]Instituto de Investigaciones para el Desarrollo Forestal (INDEFOR), Universidad de los Andes, Conjunto Forestal, 5101, Mérida, Mérida, Venezuela. [135]Departamento de Biologia, Universidade Federal do Amazonas - UFAM – Instituto de Ciências Biológicas – ICB1, Av General Rodrigo Octavio 6200, Manaus, AM 69080-900, Brazil. [136]GeoIS, El Día 369y El Telégrafo, 3° Piso, Quito, Pichincha, Ecuador. [137]Department of Ecology and Evolutionary Biology, University of Michigan, Ann Arbor, MI 48109, USA. [138]Faculty of Social Sciences, University of Nottingham, University Park, Nottingham NG7 2RD, UK. [139]Wildlife Conservation Society (WCS), 2300 Southern Boulevard, Bronx, New York, NY 10460, USA. [140]Medio Ambiente, PLUSPRETOL, Iquitos, Loreto, Peru. [141]The Mauritius Herbarium, Agricultural Services, Ministry of Agro-Industry and Food Security, Reduit 80835, Mauritius. [142]Department of Biology, Aarhus University, Building 1540, Aarhus C, Aarhus 8000, Denmark. [143]Escuela de Ciencias Forestales (ESFOR), Universidad Mayor de San Simon (UMSS), Sacta, Cochabamba, Bolivia. [144]FOMABO, Manejo Forestal en las Tierras Tropicales de Bolivia, Sacta, Cochabamba, Bolivia. [145]Departamento de Ciencias Forestales, Universidad Nacional de Colombia, Calle 64 x Cra 65, Medellín, Antioquia 1027, Colombia. [146]Fundación Amigos de la Naturaleza (FAN), Km. 7 1/2 Doble Vía La Guardia, Santa Cruz, Bolivia. [147]Tropenbos International, Horaplantsoen 12, Ede 6717 LT, The Netherlands. [148]School of Anthropology and Conservation, University of Kent, Marlowe Building, Canterbury, Kent CT2 7NR, UK. [149]Herbario Nacional del Ecuador, Universidad Técnica del Norte, Quito, Pichincha, Ecuador. [150]Instituto de Biodiversidade e Florestas, Universidade Federal do Oeste do Pará, Rua Vera Paz, Campus Tapajós, Santarém, PA 68015-110, Brazil. [151]Facultad de Biologia, Universidad Nacional de la Amazonia Peruana, Pevas 5ta cdra, Iquitos, Loreto, Peru. [152]Department of Biology, University of Missouri, St. Louis, MO 63121, USA. [153]Deceased: Dairon Cárdenas López, Cid Ferreira. ✉email: hans.tersteege@naturalis.nl

