## [Peer Review File · Communications Biology]

Reviewers' comments:

Reviewer #1 (Remarks to the Author):

General Comment

ter Steege et al. present an original piece of work entitled "Mapping tree density, alpha-diversity and species-richness of the Amazonian tree flora". The study maps species diversity at a fine scale resolution across amazonia and explores its determinants. They drew on the large Amazon Tree Diversity Network taking advantage of 2,046 plots. The major findings and contributions are that:

1. location and four major soil-forest combinations explains close to 70% of species richness spatial distribution;
2. seasonality, with water deficit, tree density, and collecting density explain 42% of species richness spatial distribution in Terra Firme;
3. they share all the plot metadata and the gridded maps at 0.1°.

The topic is timely and of primary importance to investigate the process of regional species richness distribution in Amazonia. I thank the authors as I really enjoyed reading their work and appreciate that they share the resulting data. I have no major comments but only a few minor comments that I would like to see addressed before publication but that can be ignored. In a nutshell, they include:

1. using a goodness of prediction evaluation and not only a goodness-of-fit;
2. acknowledging the log-normal distribution of diversities and richnesses values and maybe use a transformation before using them in regressions;
3. better introducing the regional density;
4. including a summary table;
5. discussing the obtained map and its goodness-of-fit against previous works such as teledetection products.

Abstract

- * "Using only location as a predictor and stratified to account for the four major soil-forest combinations": The sentence might be a bit misleading as you are thus not using location-only but location and the four major soil-forest combinations.
- * "We suggest that the size and fragmentation of these systems drives to a large extent their large-scale diversity and hence local diversity": Could you test this interpretation?
- * "CWD, tree density and, perhaps, collecting density": At first read I did not get the meaning of CWD, please avoid the acronym, and the "perhaps" collecting density is fuzzy when you have not seen the results yet.
- * "meta plot data": Do you mean plot metadata? By the way, thanks for sharing this amazing dataset.

Introduction

- * Thanks for nicely introducing the limits of your dataset. Is the contributor's effect too spatially structured to test its importance with a random effect in regressions?

Methods

- * Are white sands plots sufficiently distributed across amazonia for prediction across the region? It seems that you lack southern plots? Do you think it may affect your results? Anyway, I reckon southern white sands represent only a few pixels of your diversity and richness maps.
- * You only evaluated your model by goodness-of-fit, which is nice to evaluate the variation explained by your factors as you did. But, to my mind, your analyses lacked an evaluation of the goodness of

prediction of your model before showing projections. Why not use root mean square error of predictions or other methods to give an idea of the quality of predictions, which I think should reinforce your message?

* You only did regressions for species richness (Sha), why not report results for the diversity (alpha)?

Results

* For regression when testing forest types effect (b) and evaluating goodness-of-fit (d) in figures 1 & 2 (and corresponding SIs) I wonder if it would not be better to acknowledge the log-normal distribution of your data (diversity and richness) and use a log-normal transformation. Are the regressions diagnostic acceptable? I also think log-normal moments could be more useful on histograms (1a & 2a) as the mode and the dispersion seems more interesting to me to interpret the distribution.

* I found it nice to present diversity alpha, and richness S, Sha, and S500. But I think to gain space you might put alpha results in SIs similarly to S500, as it is the same results as Sha on which you focused to ease ecological interpretation. Or have I missed something? Anyway, keeping alpha in the main text as is, works well too, it is just a suggestion.

* I am unsure of what is regional density, to my mind it should be clearer in methods.

* There are a lot of results on the effect of co-variables (local density, regional density, collection density, CWD, rainfall, collect year), for several quantiles regressions (50th and 90th) for several response variables (diversity alpha, richnesses S, Sha, S500). The text is nice but I wonder if a summary table could not help the reader to summarise all these results at once?

* Similarly, besides relatively simple, the sentence summarising two or three variables models was a bit hard to read: "CWD 27%, Tree density (D) 24% and Collecting density (CD) 13%; combined CWD+D 38%; CWD+CI 28%; D+CI 28%. CWD+D+CI 38%".

Discussion

* I was surprised that the effect of collecting density was not clearly discussed as a bias. Did I misunderstand something?

* Could you quickly define the Acacia Mts?

* Could you discuss the choice of only seven major soil-forest combinations summarised into four? I imagine there is a balance between the number of classes and the number of plots per class. But it lacked a bit of discussion to me.

* You claim you produced the most accurate map up-to-date. I believe you but it lacks comparison with other studies. Are there possible comparisons with teledetection products? What about Saatchi et al. 2008?

References

Saatchi, S., Buermann, W., Ter Steege, H., Mori, S., & Smith, T. B. (2008). Modeling distribution of Amazonian tree species and diversity using remote sensing measurements. *Remote Sensing of Environment*, 112(5), 2000-2017.

Reviewer #2 (Remarks to the Author):

General appreciation:

I have read the manuscript ID COMMSBIO-23-1689-T and entitled "Mapping tree density, alpha-diversity and species richness of the Amazonian tree flora." The authors used an unprecedented dataset of tree occurrences and their abundances. Then using this data, the authors estimated several metrics of alpha diversity and applied a spatial model to make predictions over the entire Amazon region. The authors also assessed the effect of several environmental variables on the metrics of alpha diversity.

The study presents a good amount of work. The authors compile novel data. Their analyses are extensive, but the spatial model is poorly described. The results are interesting; however, I found myself wanting to know more about the spatial model, the impacts of using locations as predictors, the associated errors (commission and omission), and uncertainty (residuals).

Comments

1) The introduction is well-written, and the authors properly describe the goal and purpose of their paper. However, I found some conflicts regarding the terminology used in the introduction and the title of the authors' manuscript. Alpha-diversity encompasses the diversity in a sample; thus, density, and species richness are metrics of alpha-diversity. Fisher's alpha is a metric of alpha-diversity itself and not alpha-diversity on its own. I suggest the authors standardize the terminology used in the title and the text to avoid confusion with the readership.

2) Spatial interpolation model. Along the text, the authors indicate they used a spatial model; however, there is little description of the spatial interpolation model. Is it a model global interpolation or a local interpolation model? In any of the cases, what is the name of the interpolation used, and what are the parameters used? How did the authors deal with data aggregation?

Although the authors presented a good description of the metric calculations and the classification (simplification) of the soil categories, they failed to describe the modeling method used. This missing information must be clearly stated, as it is one of the major goals of this work. Moreover, in order to allow the reproducibility of the outputs, the R codes must be deposited in a public repository.

Additional information is required regarding the uncertainty of model predictions; which regions present high or low uncertainty? Areas/regions with high uncertainty may indicate that our models are not performing very well and can give us novel insights regarding the modeling procedure and the mechanisms that generate the observed patterns.

3) The discussion section is interesting; however, the authors discuss very little about the implications of the models. In other words, a section that outlines of limitations of the results and interpretation is necessary.

I hope the authors don't find my comments/suggestions offensive or aggressive, but I hope at least they find these comments helpful.

Reviewer #3 (Remarks to the Author):

Review Report

This paper tried to map tree density, alpha diversity and tree richness of the Amazon tree flora. As one of the most diverse terrestrial ecosystem, understanding its diversity and its relationship with abiotic factors seems to be interesting not only to the biologists but also to others including policy makers. The paper uses latest (2023) huge data set of more than two thousand plots from the forests of Amazonia through Amazon Tree Diversity Network (ATDN). Hence this is a great attempt to understand the diversity and dynamics of one of the rich and diverse tropical ecosystem. The study also novel in the sense no such extensive study has recently reported with such a data set. The earlier study was restricted to Terra-frime forests and it was two decade older. More importantly current study is at very higher resolution (0.1 degree) comparing to the 1 degree resolution map in 2000 and 2003. The paper claims that major soil/forest type exerts a significant and strong influence on tree diversity in Amazonia and claims upto 70% accuracy which is unprecedented. This result is statistically inverse to species dominance and that we could expect too. By analyzing rainfall pattern, the authors also claims that the annual rainfall and Cumulative Water Deficit have a strong impact on species richness of terra-frime forest. The seven main forest soil combinations have a strong effect on tree species diversity and richness. They also claim that at large scale water is a more important driver for tree species richness than is soil fertility. Using location as predictor their spatial model claims the most accurate spatial model which explaining close to 70% of the diversity and richness of Amazonian forest plots. All these claims are found to be convincing and supported by good statistical analysis and figures.

Certainly, some worries are there, especially how the species identification made error free what are keen steps that has been taken for data correction and curing for such huge dataset. The limitation in this regard to be clearly spelt out in the methodology part. For me, the paper will be more robust and remarkable if they can include the impact of factors like fire and altitudinal gradient too. Please give expanded form Cumulative Water Deficit (CWD) in the abstract. My section based comments/queries are following

Introduction

1. In the first paragraph, there is list of drivers responsible for tree alpha diversity. But I could not find forest fire which is prominent in Amazonia (Refer Feng et al, 2021 and allied papers)
2. Relative area plays a major role and small and fragmented area certainly will be low in diversity as explained in paragraph 2, but in swamp forests the niche also plays a major role since only unique species could survive there and that would be minimum too. Hence the species diversity will be poor but due to its unique nature those systems plays a major role in nature. And more importantly those species may be endemic. I would like to suggest the paper could address the endemism also. My point is that low diverse system need not always to be at lower end with respect to their conservation importance.
3. In paragraph three, it indicates that for one type of forest (Terra-firme) they have previous data set for 2003. It would be more informative to the readers, if the authors can provide a temporal change in density and diversity, at least for this forest type (I know there is difference in resolution of map, but within that, something can be think of)
4. What about the steps that have been take to minimize the bias mentioned in fifth paragraph?

Methods

5. Is there any standard definition followed by ATDN for 'Tree', if that should be mentioned
6. The methodology says that the data has been taken from 2046 plots. What about the minimum girth/dbh considered for enumeration. It should be indicated in the methodology part and hope that would be common for all 2046 plots, if not those who are in common should only be considered.
7. Soil Map has been created based on 3 data sets (Soil and Terrain database for Latin America and the Caribbean. But those database seems little bit old (2005,2007, 2011). I would suggest using the latest one, if available.

Result, Discussion and conclusion and Reference

8. What about the altitudinal gradient of the study area? It would be more impressive and informative if you can also link the impact of altitudinal gradient on density and diversity
9. Is the study area fire prone? If yes, do you have a forest fire map of the study area? It will be interesting to see the impact of fire on density and diversity, if it is a fire prone area
10. There is comparison between the year of plot establishment and richness (Figure S21D). What is its significance? What was your hypothesis in this regard and why?
11. What would be the significance of this study in conservation, management and policy level
12. How this study will lead to further studies? In other way, how these study supports to studies in near future?

I also suggest to have look at the following references and consider those, if found useful

1. Sabatini, F.M., Jiménez-Alfaro, B., Jandt, U. et al. Global patterns of vascular plant alpha diversity. *Nat Commun* 13, 4683 (2022). <https://doi.org/10.1038/s41467-022-32063-z>
2. https://escholarship.org/content/qt9qw3j2v2/qt9qw3j2v2_noSplash_11a25358fb8f3bfe9e4741121e35b923.pdf?t=pga0p4
3. https://www.nature.com/articles/s41586-021-03876-7.epdf?sharing_token=Hpmya-yEyOPfLsmVOeFITdRgN0jAjWel9jnR3ZoTv0NTCnBBOD2EQsyUPfbeUxYDKjya6Vzpx-r5NkL3xvPS4cpAKLxgGB5W07hikKhNYK9MVga2IZOwVpjpdnUCazDjUHMBT4KT2BxB6I47f2uMZg-T8EHmjn3KycJV-x3EIKhObk4tkbKL2aNqYJt7A3k9LsgFT9DBTXK4z8uk1ITnrswFKJ130bg57462dkhYQL6cwQ1n17oJg-E-ujhXBHe88B4g0zTzAazfzQtsVPx-

QStEHj_ZcKeLNFx4oXz2PTBDqYqMzIH03pN1Zw_RVUeVKdnJnbxkRVXmEE7URRpKZh2rSeMga6PFhmBK
3RwmM4Q=&tracking_referrer=news.mongabay.com

Reviewers' comments:

Reviewer #1 (Remarks to the Author):

General Comment

ter Steege et al. present an original piece of work entitled "Mapping tree density, alpha-diversity and species-richness of the Amazonian tree flora". The study maps species diversity at a fine scale resolution across Amazonia and explores its determinants. They drew on the large Amazon Tree Diversity Network taking advantage of 2,046 plots. The major findings and contributions are that:

1. location and four major soil-forest combinations explain close to 70% of species richness spatial distribution;
2. seasonality, with water deficit, tree density, and collecting density explain 42% of species richness spatial distribution in Terra Firme;
3. they share all the plot metadata and the gridded maps at 0.1°.

The topic is timely and of primary importance to investigate the process of regional species richness distribution in Amazonia. I thank the authors as I really enjoyed reading their work and appreciate that they share the resulting data.

Thank you for your positive feedback.

I have no major comments but only a few minor comments that I would like to see addressed before publication but that can be ignored. In a nutshell, they include:

1. using a goodness of prediction evaluation and not only a goodness-of-fit;
2. acknowledging the log-normal distribution of diversities and richness values and maybe use a transformation before using them in regressions;
3. better introducing the regional density;
4. including a summary table;
5. discussing the obtained map and its goodness-of-fit against previous works such as teledetection products.

See our comments below...

Abstract

* “Using only location as a predictor and stratified to account for the four major soil-forest combinations”: The sentence might be a bit misleading as you are thus not using location-only but location and the four major soil-forest combinations.

But this is exactly what we do, we stratify by forest type and then use only location as predictor. We brought the stratification a bit closer to location.

The abstract is changed substantially and then shortened to keep it at the 200 words limit

* “We suggest that the size and fragmentation of these systems drives to a large extent their large-scale diversity and hence local diversity”: Could you test this interpretation?

It had been tested in previous paper that we cite but we have now included a new Figure S30, to support this claim better.

* “CWD, tree density and, perhaps, collecting density”: At first read I did not get the meaning of CWD, please avoid the acronym, and the “perhaps” collecting density is fuzzy when you have not seen the results yet.

We have tried to keep acronyms to a minimum now. Collecting density is now collecting intensity

* “meta plot data”: Do you mean plot metadata? By the way, thanks for sharing this amazing dataset.

Yes, corrected

Introduction

* Thanks for nicely introducing the limits of your dataset. Is the contributor's effect too spatially structured to test its importance with a random effect in regressions?

Perhaps. Only two areas are strongly over-collected. In the regressions its effect is small. By the eye, the high diversity on Manaus and French Guiana are striking but the areas may be too small to make the effect apparent. We have added a bit more discussion on this topic. Lines 281...

Methods

* Are white sands plots sufficiently distributed across Amazonia for prediction across the region? It seems that you lack southern plots? Do you think it may affect your results? Anyway, I reckon southern white sands represent only a few pixels of your diversity and richness maps.

That is correct. White sands are mainly found in the Guianas and Central to Northern Brazilian Amazon. The loess is not allowed to extrapolate, however, so for the very far away (small) white sand patches no information is available. We made that clearer in the methods. L397 - 410

* You only evaluated your model by goodness-of-fit, which is nice to evaluate the variation explained by your factors as you did. But, to my mind, your analyses lacked an evaluation of the goodness of prediction of your model before showing projections. Why not use root mean square error of predictions or other methods to give an idea of the quality of predictions, which I think should reinforce your message?

We did a **goodness of fit** of the predictions via a regression, indeed. What we also provided were maps of the residuals and boxplots of the residuals vs region and forest type. In neither of the cases any spatial structure in the residuals and very little information was left. Hence both in the prediction and residuals the mapping performs quite well. What we have added now is the **goodness of prediction** (as requested) by adding a map of the standard error of the loess regression and analyzing it by forest type and region. Indeed, this provided extra insight in the effect by forest type. Thank you for this suggestion. L159 – 162 and Figure S7; L185-189 and Figure S11

* You only did regressions for species richness (Sha), why not report results for the diversity (alpha)?

We did similar analyses for tree alpha-diversity (Fisher's alpha), tree species-richness / ha and species richness / 500 ind. We have added the goodness of prediction for each of these as well now. Figures S14-16

Only the main results of tree alpha-diversity and tree species-richness/ha are in the main text, the other material is in the supplement.

Results

* For regression when testing forest types effect (b) and evaluating goodness-of-fit (d) in figures 1 & 2 (and corresponding SIs) I wonder if it would not be better to acknowledge the log-normal distribution of your data (diversity and richness) and use a log-normal transformation. Are the regressions diagnostic acceptable? I also think log-normal moments could be more useful on histograms (1a & 2a) as the mode and the dispersion seems more interesting to me to interpret the distribution.

L165-168 ; 189-190: We have added a mapping for $\log(\text{Fisher's alpha})$ (Figure S9) and $\log(\text{species-richness})$ (Figure S13). In each case the prediction of the map is somewhat better. We prefer to produce the maps in our manuscript with the actual values, as the number of species per ha is easier to interpret than the log of it.

* I found it nice to present diversity alpha, and richness S, Sha, and S500. But I think to gain space you might put alpha results in SIs similarly to S500, as it is the same results as Sha on which you focused to ease ecological interpretation. Or have I missed something? Anyway, keeping alpha in the main text as is, works well too, it is just a suggestion.

As Fisher's alpha is a widely used metric and our previous maps used that, we prefer to keep both results in the main text.

* I am unsure of what is regional density, to my mind it should be clearer in methods.

It is the interpolated density (loess model result), based on the plot data. We have added more explanation in the methods section. L420-425

* There are a lot of results on the effect of co-variables (local density, regional density, collection density, CWD, rainfall, collect year), for several quantiles regressions (50th and 90th) for several response variables (diversity alpha, richnesses S, Sha, S500). The text is nice but I wonder if a summary table could not help the reader to summarise all these results at once?

We have added a supplementary table S1 for the effects on species richness

* Similarly, besides relatively simple, the sentence summarising two or three variables models was a bit hard to read: "CWD 27%, Tree density (D) 24% and Collecting density (CD) 13%; combined CWD+D 38%; CWD+CI 28%; D+CI 28%. CWD+D+CI 38%".

These have also been added to the table S1

Discussion

* I was surprised that the effect of collecting density was not clearly discussed as a bias. Did I misunderstand something?

True. We have added a little more discussion. L281 - 290

* Could you quickly define the Acarai Mts?

Done, L311

* Could you discuss the choice of only seven major soil-forest combinations summarised into four? I imagine there is a balance between the number of classes and the number of plots per class. But it lacked a bit of discussion to me.

We have added that to the methods. L380 - 388

* You claim you produced the most accurate map up-to-date. I believe you but it lacks comparison with other studies. Are there possible comparisons with teledetection products? What about Saatchi et al. 2008?

One of us is co-author of that paper. The map (Fig 6, which combines the plots with tele-detection) is based on a much smaller sample of plots, uses Maxent to map diversity in different classes and produces high diversity estimates where no such plots have been found, whereas it predicts rather low diversity, where our data shows the highest. It was an interesting experiment to use maxent to map diversity in different classes and combine them into one map but the result is not so good - with hindsight.

References

Saatchi, S., Buermann, W., Ter Steege, H., Mori, S., & Smith, T. B. (2008). Modeling distribution of Amazonian tree species and diversity using remote sensing measurements. *Remote Sensing of Environment*, 112(5), 2000-2017.

Reviewer #2 (Remarks to the Author):

General appreciation:

I have read the manuscript ID COMMSBIO-23-1689-T and entitled "Mapping three density, alpha-diversity and species richness of the Amazonian tree flora." The authors used an unprecedented dataset of tree occurrences and their abundances. Then using this data, the authors estimated several metrics of alpha diversity and applied a spatial model to make predictions over the entire Amazon region. The authors also assessed the effect of several environmental variables on the metrics of alpha diversity.

The study presents a good amount of work. The authors compile novel data. Their analyses are extensive, but the spatial model is poorly described.

We tried to improve the description. L389 - 410

The results are interesting; however, I found myself wanting to know more about the spatial model, the impacts of using locations as predictors, the associated errors (commission and omission), and uncertainty (residuals).

Based on the comments of R1, we have now also included a map with standard errors of the model. Figures S7, S11

Comments

1) The introduction is well-written, and the authors properly describe the goal and purpose of their paper. However, I found some conflicts regarding the terminology used in the introduction and the title of the authors' manuscript. Alpha-diversity encompasses the diversity in a sample; thus, density, and species richness are metrics of alpha-diversity. Fisher's alpha is a metric of alpha-diversity itself and not alpha-diversity on its own. I suggest the authors standardize the terminology used in the title and the text to avoid confusion with the readership.

We have improved the wording. Tree density, is not, however, a diversity measure but simply the number of trees on a plot. We only Fisher's alpha as a measure of tree alpha-diversity and tree species-richness, for the number of species. We added the following text "*Tree alpha-diversity (here defined Fisher's alpha of a tree-inventory plot)*", and also "*Fisher's alpha provides information on both species-richness in a sample of known size and the relative abundances of all species in that sample, providing both aspects of biodiversity. Species-richness is an important aspect of biodiversity and in many cases easier to communicate. In this paper we will use both indices.*" L82-90

2) Spatial interpolation model. Along the text, the authors indicate they used a spatial model; however, there is little description of the spatial interpolation model. Is it a model global interpolation or a local interpolation model? In any of the cases, what is the name of the interpolation used, and what are the parameters used? How did the authors deal with data aggregation?

We have tried to improve the text on the interpolation but it is not much more than a spatial regression with latitude, longitude and their interaction as predicting parameters. We are not sure what is meant by data aggregation. We did not aggregate data ourselves. If clustering of data points is meant here, this is not a problem with the method used, it basically ensures a better local fit. L389-410

Although the authors presented a good description of the metric calculations and the classification (simplification) of the soil categories, they failed to describe the modeling method used. This missing information must be clearly stated, as it is one of the major goals of this work. Moreover, in order to allow the reproducibility of the outputs, the R codes must be deposited in a public repository.

We have done this and hope it is sufficient. The R code has been deposited to figshare

Additional information is required regarding the uncertainty of model predictions; which regions present high or low uncertainty? Areas/regions with high uncertainty may indicate that our models are not performing very well and can give us novel insights regarding the modeling procedure and the mechanisms that generate the observed patterns.

See the comments of R1. We have now added maps with the standard error of the loess models. L159 – 162 and Figure S7; L185-189 and Figure S11

3) The discussion section is interesting; however, the authors discuss very little about the implications of the models. In other words, a section that outlines of limitations of the results and interpretation is necessary.

We have added a bit to the discussion. L332-342

I hope the authors don't find my comments/suggestions offensive or aggressive, but I hope at least they find these comments helpful.

Not at all. Thank you for your comments.

Reviewer #3 (Remarks to the Author):

Review Report

This paper tried to map tree density, alpha diversity and tree richness of the Amazon tree flora. As one of the most diverse terrestrial ecosystem, understanding its diversity and its relationship with abiotic factors seems to be interesting not only to the biologists but also to others including policy makers. The paper uses latest (2023) huge data set of more than two thousand plots from the forests of Amazonia through Amazon Tree Diversity Network (ATDN). Hence this is a great attempt to understand the diversity and dynamics of one of the rich and diverse tropical ecosystem. The study also novel in the sense no such extensive study has recently reported with such a data set. The earlier study was restricted to Terra-firme forests and it was two decade older. More importantly current study is at very higher resolution (0.1 degree) comparing to the 1 degree resolution map in 2000 and 2003. The paper claims that major soil/forest type exerts a significant and strong influence on tree diversity in Amazonia and claims upto 70% accuracy which is unprecedented. This result is statistically inverse to species dominance and that we could expect too. By analyzing rainfall pattern, the authors also claims that the annual rainfall and Cumulative Water Deficit have a strong impact on species richness of terra-firme forest. The seven main forest soil combinations have a strong effect on tree species diversity and richness. They also claim that at large scale water is a more important driver for tree species richness than is soil fertility. Using location as predictor their spatial model claims the most accurate spatial model which explaining close to 70% of the diversity and richness of Amazonian forest plots. All these claims are found to be convincing and supported by good statistical analysis and figures.

Certainly, some worries are there, especially how the species identification made error free what are keen steps that has been taken for data correction and curing for such huge dataset. The limitation in this regard to be clearly spelt out in the methodology part. For me, the paper will be more robust and remarkable if they can include the impact of factors like fire and altitudinal gradient too.

We cannot check all species identifications but do not need the actual names for the species as no comparison between plots is made based on composition. Here, we used all morpho-species identified at the plots. We added a little more discussion on this point. L286-290

We model diversity for undisturbed plots only. Secondary forest plots, logged forest and burnt forest were not included.

Our database does not include plots over 500m, so elevation is not likely to play a major role.

Please give expanded form Cumulative Water Deficit (CWD) in the abstract.

We have done that and use the full wording across the paper

My section based comments/queries are following

Introduction

1. In the first paragraph, there is list of drivers responsible for tree alpha diversity. But I could not find forest fire which is prominent in Amazonia (Refer Feng et al, 2021 and allied papers).

ATDN only works with undisturbed plots. Some plots may have older influence of pre-Colombian settlements but recently disturbed forests (burning, logging, secondary forest) are not included.

2. Relative area plays a major role and small and fragmented area certainly will be low in diversity as explained in paragraph 2, but in swamp forests the niche also plays a major role since only unique species could survive there and that would be minimum too. Hence the species diversity will be poor but due to its unique nature those systems plays a major role in nature. And more importantly those species may be endemic. I would like to suggest the paper could address the endemism also. My point is that low diverse system need not always to be at lower end with respect to their conservation importance.

As we are not using any data on composition, we cannot make any statements on endemism. We have done that in previous and forthcoming papers but disagree with the reviewer on the mechanism explaining patterns of diversity. If all of Amazonia was either white sand forest or swamp, undoubtedly there would be far more species in these systems (the perhaps, the low diversity of terra-firme would be thought to be caused by it's high Al content (ter Steege et al 2001, JTE)). It is true that adaptation is an important aspect of low diversity of rare systems as it allows the species to be common there but the drawback is that they cannot grow well in the other systems.

3. In paragraph three, it indicates that for one type of forest (Terra-firme) they have previous data set for 2003. It would be more informative to the readers, if the authors can provide a temporal change in density and diversity, at least for this forest type (I know there is difference in resolution of map, but within that, something can be think of)

In ATDN we have only one inventory for each plot, so temporal changes are not possible with this data. New data (new plots) were simply added to the plots that were in our database first.

4. What about the steps that have been take to minimize the bias mentioned in fifth paragraph?

We cannot deal with errors in the identification, except for leaving out plots with too few properly identified species (which is the case). We assume that if it happens across Amazonia it may have an effect on absolute values (in both directions) but it will not change the pattern much. We have added this to the introduction and also added a bit more discussion on this point. L120-137

Methods

5. Is there any standard definition followed by ATDN for 'Tree', if that should be mentioned

We have added our definition used. L359-362

6. The methodology says that the data has been taken from 2046 plots. What about the minimum girth/dbh considered for enumeration. It should be indicated in the methodology part and hope that would be common for all 2046 plots, if not those who are in common should only be considered.

This would be 10 cm dbh – we have added this to the methods section L359-362

7. Soil Map has been created based on 3 data sets (Soil and Terrain database for Latin America and the Caribbean. But those database seems little bit old (2005,2007, 2011). I would suggest using the latest one, if available.

For soil types the map of Quesada (Quesada et al 2011 Soils of Amazonia with particular reference to the RAINFOR sites BioGeoSciences) is probably the best interpretation of soil types in Amazonia at this moment. For soil fertility we have now used the latest version of the map, Zuquim has been working on for a long time (Zuquim et al 2023 Introducing a map of soil base cation concentration, an ecologically relevant GIS-layer for Amazonian forests). Hence, we now only use pH and sum of bases. L380-388, 426-435

Result, Discussion and conclusion and Reference

8. What about the altitudinal gradient of the study area? It would be more impressive and informative if you can also link the impact of altitudinal gradient on density and diversity

We have no plots over 500m, so altitude is not likely playing an important role

9. Is the study area fire prone? If yes, do you have a forest fire map of the study area? It will be interesting to see the impact of fire on density and diversity, if it is a fire prone area

We have only used undisturbed plots.

10. There is comparison between the year of plot establishment and richness (Figure S21D). What is its significance? What was your hypothesis in this regard and why?

Our text on this point is *“Plots of the ATDN were established between 1934 and 2020. The number of known (**described**) species for Amazonia has **increased considerably** in that period, especially since the 1940s. Also, new taxonomic insights contribute to splitting of dominant taxa into separated lineages. Thus, it should be expected that, all other being equal, early plots have fewer species than plots established more recently.”*

We have also added *“The effect of this would depend on the quality of the botanists identifying the species, as they could either force a tree into a known species or keep it as a morpho species, in which case there would be no effect on tree alpha-diversity and tree species-richness.”*.

11. What would be the significance of this study in conservation, management and policy level

We have added a small paragraph. L332-342

12. How this study will lead to further studies? In other way, how these study supports to studies in near future?

We did not add text on this aspect but as we provide all data necessary for the maps, anyone can use the data for his/her purposes.

I also suggest to have look at the following references and consider those, if found useful

1. Sabatini, F.M., Jiménez-Alfaro, B., Jandt, U. et al. Global patterns of vascular plant alpha diversity. Nat Commun 13, 4683 (2022). <https://doi.org/10.1038/s41467-022-32063-z>

2.

https://escholarship.org/content/qt9qw3j2v2/qt9qw3j2v2_noSplash_11a25358fb8f3bfe9e4741121e35b923.pdf?t=pga0p4

The map reproduced in this paper is in fact our own 1-degree map of 2003.

3. https://www.nature.com/articles/s41586-021-03876-7.epdf?sharing_token=Hpmya-yEyOPfLsmVOeFITdRgN0jAjWel9jnR3ZoTv0NTCnBBOD2EQsyUPfbeUxYDKjya6Vzpx-r5NkL3xvPS4cpAKLxgGB5W07hikKhNYK9MVga2IZOwVpjpdnUCazDjUHMBT4KT2BxB6I47f2uMZg-T8EHmjn3KycJV-x3EIKhObk4tkbKL2aNqYJt7A3k9LsgFT9DBTXK4z8uk1ITnrswFkJ130bg57462dkhYQL6cwQ1n17oJg-E-ujhXBHe88B4g0zTzAazfzQtsVPx-QStEHj_ZcKeLNFx4oXz2PTBDqYqMzIH03pN1Zw_RVUeVKdnJnbxkRVXmEE7URRpKZh2rSeMga6PFhmBK3RwmM4Q=&tracking_referrer=news.mongabay.com

We have included these references in the beginning of the discussion, plus a more recent one (also global, Liang et al 2022) to the discussion. L246-254

The species richness map in Feng is based on all plants(?) but its pattern shows little similarity with ours based on plots and based on tree collection data (Gomes et al 2019 Amazonian tree species threatened by deforestation and climate change – image below).

Feng

Gomes

Reviewers' comments:

Reviewer #1 (Remarks to the Author):

I am pleased with the way ter Steege et al. have responded to my previous comments, and I recommend publication of the manuscript. There are two points that I would have liked to see further explored, but I don't think they should override acceptance, and they could only be incorporated if a new revision cycle took place.

As far as the logarithmic transformation is concerned, I don't understand why keep the natural model with inferior fit quality. The authors claim that natural variations are easier to read. I completely agree, but the projections could be back-transformed to make maps and graphs easier to read while using a logarithmic scale and logarithmic response variable for a better fit and better visualization of regions with lower values that the high values hide in the current maps.

As far as the quality of the prediction is concerned, I only partly agree with the authors' answer. It is indeed more convincing now, but it would be preferable if the evaluation data set were not the same as the training data set, using k-folds or leave-one-out approaches, for example.

Finally, the terms "collection density" and "collection intensity" have both remained in the manuscript.

However, despite these minor comments, I thoroughly enjoyed this manuscript and recommend its publication.

Reviewer #3 (Remarks to the Author):

I have gone through both at the manuscript and the feedback from authors. It is found that the suggestions/comments are well taken and subsequent modifications has been done in respective part of the manuscript. In current form the manuscript found to be novel and scientifically tested and high quality work. Congratulations to the authors for their excellent work and best wishes to them

Reviewer #4 (Remarks to the Author):

The manuscript COMMSBIO-23-1689A provides a comprehensive analysis of local forest tree species diversity, in terms of Fisher's Alpha and per-hectare species richness, across Amazonia. The analysis consists of a series of multiple-regression models and spatial statistics, based on 2,046 in situ forest inventory plots. The manuscript further addressed potential mechanisms behind the estimated biodiversity pattern by assessing the contribution of soil, hydrology, tree-density, and bioclimate.

I found this study very timely, and the output wall-to-wall map of tree species diversity is of immense value to forest conservation and restoration. The dataset of 2,046 ATDN sample plots will be extremely useful to a large number of ongoing and upcoming research studies on Amazon rainforests. The manuscript is well-structured and easy-to-read. That being said, I have a few suggestions to further improve this manuscript.

A) The context. Although there are numerous studies addressing the biodiversity patterns of Amazonia (including Refs 37-40 in this paper), only Ref #38 addresses the pattern of local tree species richness that is directly comparable to this study. However, this manuscript only mentioned the reference in one short sentence (L.251- "Other, global, mapping exercises have also included Amazonia but with far less plot data, providing a much courser pattern 37,38"). The problems are two-fold. First, it is not correct that the global maps are "courser" in pattern, as Ref #38 provides a map in ~3km resolution,

which is even higher than this manuscript (0.1 degree). Second, an in-depth discussion will be helpful on the overall consistency in the biodiversity patterns in this study and Ref #38, as well as the subtle differences between the two. For instance, this study discovered that the highest per-hectare species richness occurred in Central Amazonia (Fig.3B), whereas Ref #38 pinpointed north-western Amazonia. Is this because of more data points that are available in Central Amazonia for this study? A comparison and discussion will further highlight the value of this study.

B) Details are lacking for the spatial models that were developed to interpolate point data (Figs 1,2) to the entire Amazonia (Fig 3). What is the nature of the models (multiple regression? linear or polynomial? with or without spatial variogram in the residuals?)? What are the coefficients of these models? Showing the partial dependence of tree species diversity values on the top predictors will be very helpful as well.

C) Results section – Figures cited in the Supplementary Information are not in order.

D) L. 184 “its residuals showed no significant spatial autocorrelation (Moran’s $I < 0.001$ n.s.)” Moran’s I is an empirical measure that indicates the magnitude of spatial autocorrelation, whereas statistical significance (as in its residuals showed no significant spatial autocorrelation) can only be derived from the p-value. To fix, you can either change “significant” to “obvious/conspicuous,” or add a p-value.

E) L.296 – “therefor” should be “therefore”

F) L.346 – “Our spatial model provides the most accurate map of tree diversity in Amazonia to date, explaining close to 70% of variation...” The statement is problematic, as R^2 value is not a direct indicator of accuracy. To support the claim that this is “the most accurate map of tree diversity in Amazonia to date,” you will need to compare the spatial interpolation results here against those from other studies (e.g. Ref #38), in a post-sample validation or cross-validation.

G) L.429 – It’s exciting to see that Amazonia-wide soil maps are capable of representing fine-scale soil variations, whereas a global soil database (e.g. WISE) is rarely shown to be useful in predicting large-scale biodiversity patterns (e.g. Ref #38). It would be worthwhile to discuss this in the general discussion that I suggested in Comment (A).

We therefore invite you to revise and resubmit your manuscript, taking into account the points raised. Please address the remaining concerns of Reviewer #1 as well as the concerns of Reviewer #4 in your revised manuscript.

I have highlighted all changes in the text in yellow. I have also made minor changes (tree-density to tree density and collecting density to collecting intensity), that I have not highlighted.

Reviewer #1 (Remarks to the Author):

I am pleased with the way ter Steege et al. have responded to my previous comments, and I recommend publication of the manuscript. There are two points that I would have liked to see further explored, but I don't think they should override acceptance, and they could only be incorporated if a new revision cycle took place.

Thank you for these comments. As there is a new review round, we will answer your comments.

As far as the logarithmic transformation is concerned, I don't understand why keep the natural model with inferior fit quality. The authors claim that natural variations are easier to read. I completely agree, but the projections could be back-transformed to make maps and graphs easier to read while using a logarithmic scale and logarithmic response variable for a better fit and better visualization of regions with lower values that the high values hide in the current maps.

For Fisher's alpha, the fit is indeed about 6% lower (66% vs 72%). For species richness, with a much less skewed distribution, the difference is marginal (72% vs 71%). We understand the problem with very high (and low values) and deal with them in our maps by truncating the values for the maps at mean plus and minus 2SD. The result is shown below. This is also explained in the legend of the figures. Below we show three maps

Left is the map for $^{10}\log(\text{Fisher's alpha})$ with all values being used for mapping; middle is the map with all data as is, the map is strongly influenced by data smaller than mean $- 2*SD$ and mean $+ 2*SD$ (technically about 5% of the data); finally the right map is the final map with truncated data, as we have used in the paper. We believe this map shows the most informative pattern. This is explained in the legend of the figures. If we model with $^{10}\log(\text{alpha})$ and back transform it, the regression would be a bit better, but the high values still need to be truncated and the map is identical. We hope the reviewer agrees that this is a suitable solution. For species richness, probably

the maps that is going to be used the most, the differences are (as said above) marginal. We hope the reviewer can live with our choice.

As far as the quality of the prediction is concerned, I only partly agree with the author's answer. It is indeed more convincing now, but it would be preferable if the evaluation data set were not the same as the training data set, using k-folds or leave-one-out approaches, for example.

Using a training set and a prediction set with this data would severely limit the predictions as the power comes from the number of samples included. We do appreciate the leave-one-out suggestion and have added this to the results. As requested by the reviewer, we produced 2046 leave-one-out maps, using that model to predict the richness of the plot left-out. The samples of white sand forest and swamp forest are much smaller than that of terra firme and varzea and there is a considerable difference. The explained variation for the total data however drops from 71% to 65%, so the plots have some influence in their own estimate. Lines 441-442 and 189-196

I have added the code to the script file.

Finally, the terms "collection density" and "collection intensity" have both remained in the manuscript.

Apologies, that has been corrected now.

However, despite these minor comments, I thoroughly enjoyed this manuscript and recommend its publication.

Thank you. We hope you find the manuscript even a bit better now.

Reviewer #3 (Remarks to the Author):

I have gone through both at the manuscript and the feedback from authors. It is found that the suggestions/comments are well taken and subsequent modifications has been done in respective part of the manuscript. In current form the manuscript found to be novel and scientifically tested and high quality work. Congratulations to the authors for their excellent work and best wishes to them

Thank you.

Reviewer #4 (Remarks to the Author):

The manuscript COMMSBIO-23-1689A provides a comprehensive analysis of local forest tree species diversity, in terms of Fisher's Alpha and per-hectare species richness, across Amazonia. The analysis consists of a series of multiple-regression models and spatial statistics, based on 2,046 in situ forest inventory plots. The manuscript further addressed potential mechanisms behind the estimated biodiversity pattern by assessing the contribution of soil, hydrology, tree-density, and bioclimate.

I found this study very timely, and the output wall-to-wall map of tree species diversity is of immense value to forest conservation and restoration. The dataset of 2,046 ATDN sample plots will be extremely useful to a large number of ongoing and upcoming research studies on Amazon rainforests. The manuscript is well-structured and easy-to-read. That being said, I have a few suggestions to further improve this manuscript.

Thank you for your comments. We are happy you appreciate the work.

A) The context. Although there are numerous studies addressing the biodiversity patterns of Amazonia (including Refs 37-40 in this paper), only Ref #38 addresses the pattern of local tree species richness that is directly comparable to this study. However, this manuscript only mentioned the reference in one short sentence (L.251- "Other, global, mapping exercises have also included Amazonia but with far less plot data, providing a much coarser pattern 37,38"). The problems are two-fold. First, it is not correct that the global maps are "coarser" in pattern, as Ref #38 provides a map in ~3km resolution, which is even higher than this manuscript (0.1 degree).

This is correct. The reviewer may have noted that several of us are also co-author on that paper. What we mean is that, although the map has a resolution that is finer than our maps here the general pattern that emerges is more coarse and ignores the riverine and white sand areas. See further below.

Second, an in-depth discussion will be helpful on the overall consistency in the biodiversity patterns in this study and Ref #38, as well as the subtle differences between the two. For instance, this study discovered that the highest per-hectare species richness occurred in Central Amazonia (Fig.3B), whereas Ref #38 pinpointed north-western Amazonia. Is this because of more data points that are available in Central Amazonia for this study? A comparison and discussion will further highlight the value of this study.

We have added this to the discussion. Lines 254-273, 352-354

B) Details are lacking for the spatial models that were developed to interpolate point data (Figs 1,2) to the entire Amazonia (Fig 3). What is the nature of the models (multiple regression? linear or polynomial? with or without spatial variogram in the residuals)? What are the coefficients of these models? Showing the partial dependence of tree species diversity values on the top predictors will be very helpful as well.

We have used loess regression, as explained in the methods lines 412-446 with a graphical description in figure S29. As mentioned in our methods, we did not use Kriging, because most variation is locally present (we know this as on several sites we have many plots), hence there is no range and Kriging will become inverse distance weighting. As this model does not have an error function, we have used Loess regression.

C) Results section – Figures cited in the Supplementary Information are not in order.

Correct, they were still in the order when the methods section was after the introduction. This has been corrected.

D) L. 184 “its residuals showed no significant spatial autocorrelation (Moran’s I < 0.001 n.s.)”
Moran’s I is an empirical measure that indicates the magnitude of spatial autocorrelation, whereas statistical significance (as in its residuals showed no significant spatial autocorrelation) can only be derived from the p-value. To fix, you can either change “significant” to “obvious/conspicuous,” or add a p-value.

We have tested the significance with the function Moran.I() from the library *ape*, that calculates the p value. This is the value that is given in the graph. We have added this to the methods (line 500) and the legends of the figures. Line 441

E) L.296 – “therefor” should be “therefore”

Corrected

F) L.346 – “Our spatial model provides the most accurate map of tree diversity in Amazonia to date, explaining close to 70% of variation...” The statement is problematic, as R2 value is not a direct indicator of accuracy. To support the claim that this is “the most accurate map of tree diversity in Amazonia to date,” you will need to compare the spatial interpolation results here against those from other studies (e.g. Ref #38), in a post-sample validation or cross-validation.

We have done that now, see above.

G) L.429 – It’s exciting to see that Amazonia-wide soil maps are capable of representing fine-scale soil variations, whereas a global soil database (e.g. WISE) is rarely shown to be useful in predicting large-scale biodiversity patterns (e.g. Ref #38). It would be worthwhile to discuss this in the general discussion that I suggested in Comment (A).

The reason for this is that we use soil type here as a discrete parameter and map for each soil type separately, without using fertility or acidity or any other soil parameter. In other instances, researchers have used the soil parameter values, such as pH, N, sum of bases, CEC, which have a lot of local variation. When such data are interpolated (as we did for pH and Zuquim et al for SB), they have a rather smooth pattern, high error and noticed that if we did such interpolation and assigned the map values to each soil pit location, the sum of bases of várzea forest is systematically underestimated, because the interpolation uses also the soil data of nutrient poor terra firme forest close by – see figures below.

We only used soil parameter data (sum of bases, as there was a brand new map layer Zuquim et al) and pH (as there are very many data points for pH) to test the effect on richness of TF as this forest type has the largest amount of plots and most soils pits are in terra firme forest. We made a small comment of this in our discussion of the different maps.

REVIEWERS' COMMENTS:

Reviewer #1 (Remarks to the Author):

I am once again very pleased with the way ter Steege et al. have responded to my previous comments, and I again recommend publication of the manuscript. I will certainly live with their choice and am more than satisfied with the result. I'm also more interested in the message than in the small differences in fit. But I find that the new truncated map shows much more information now. Thanks for taking the time to implement the leave-one-out strategy. If I may say so, the only limitation I can see is that since the model is primarily spatial, excluding a single sample doesn't seem to be the most conservative evaluation strategy. However, I understand your point about predictive power. Once again, I congratulate the authors on their patience and wish them that this is the last evaluation cycle!

Reviewer #4 (Remarks to the Author):

The manuscript COMMSBIO-23-1689B has undergone significant enhancements, and I am delighted to observe that my feedback has made a valuable contribution.

I have just two minor suggestions, both of which pertain to the accessibility of data and code.

Firstly, as per the rebuttal letter, the "loess regression" technique was employed for mapping, as detailed in the methods section (lines 412-446) with visual clarification in figure S29. To ensure the reproducibility of the results, it would be advisable for the authors to furnish all model coefficients and related parameters in a structured tabular format. Alternatively, the authors might consider sharing the model on platforms like Figshare or GitHub to facilitate evaluation and use by others.

Secondly, the data availability statement includes a link to plot-level metadata, but unfortunately, this link is inaccessible. Moreover, there is a lack of clarity regarding the means by which the audience can access tree-level data. While we understand that certain tree-level data may be subject to proprietary or confidential constraints, it would be greatly beneficial if the authors could explicitly acknowledge such restrictions. Furthermore, providing a clear pathway for interested parties to obtain such data, such as by contacting the corresponding author or a designated data committee, would be immensely valuable to the scientific community.

REVIEWERS' COMMENTS:

Reviewer #1 (Remarks to the Author):

I am once again very pleased with the way ter Steege et al. have responded to my previous comments, and I again recommend publication of the manuscript. I will certainly live with their choice and am more than satisfied with the result. I'm also more interested in the message than in the small differences in fit. But I find that the new truncated map shows much more information now. Thanks for taking the time to implement the leave-one-out strategy. If I may say so, the only limitation I can see is that since the model is primarily spatial, excluding a single sample doesn't seem to be the most conservative evaluation strategy. However, I understand your point about predictive power. Once again, I congratulate the authors on their patience and wish them that this is the last evaluation cycle!

Thank you. Hope indeed that you enjoy reading the final version

Reviewer #4 (Remarks to the Author):

The manuscript COMMSBIO-23-1689B has undergone significant enhancements, and I am delighted to observe that my feedback has made a valuable contribution.

Thank you and also thank you for your comments

I have just two minor suggestions, both of which pertain to the accessibility of data and code.

Firstly, as per the rebuttal letter, the "loess regression" technique was employed for mapping, as detailed in the methods section (lines 412-446) with visual clarification in figure S29. To ensure the reproducibility of the results, it would be advisable for the authors to furnish all model coefficients and related parameters in a structured tabular format. Alternatively, the authors might consider sharing the model on platforms like Figshare or GitHub to facilitate evaluation and use by others.

This is exactly what has happened. All data and source code has been available on FigShare all the time. Due to transfer from one NPG journal to Communications Biology, this link did not work anymore. This was out of my hands.

Secondly, the data availability statement includes a link to plot-level metadata, but unfortunately, this link is inaccessible.

See above

Moreover, there is a lack of clarity regarding the means by which the audience can access tree-level data. While we understand that certain tree-level data may be subject to proprietary or confidential constraints, it would be greatly beneficial if the authors could explicitly acknowledge such restrictions. Furthermore, providing a clear pathway for interested parties to obtain such data, such as by contacting the corresponding author or a designated data committee, would be immensely valuable to the scientific community.

Tree by tree data is not part of the ATDN database and that data remains with the separate data owners. It is possible to get access to the species lists in ATDN upon reasonable request but as this data is not needed to repeat the analyses here, we have opted not to add this to the data availability. All data needed to repeat this work is provided in the online data